# Highly specific and non-invasive imaging of Piezo1-dependent activity across scales using GenEPi

Sine Yaganoglu[1,8], Konstantinos Kalyviotis [2,8], Christina Vagena-Pantoula [2], Dörthe Jülich[3], Benjamin M. Gaub[1], Maaike Welling[1,2], Tatiana Lopes[4], Dariusz Lachowski [2], See Swee Tang[2], Armando Del Rio Hernandez [2], Victoria Salem [2], Daniel J. Müller [1], Scott A. Holley[3], Julien Vermot[2], Jian Shi[5], Nordine Helassa [6,7], Katalin Török [6] & Periklis Pantazis [1,2] ✉

Mechanosensing is a ubiquitous process to translate external mechanical stimuli into biological responses. Piezo1 ion channels are directly gated by mechanical forces and play an essential role in cellular mechanotransduction. However, readouts of Piezo1 activity are mainly examined by invasive or indirect techniques, such as electrophysiological analyses and cytosolic calcium imaging. Here, we introduce GenEPi, a genetically-encoded fluorescent reporter for non-invasive optical monitoring of Piezo1-dependent activity. We demonstrate that GenEPi has high spatiotemporal resolution for Piezo1-dependent stimuli from the single-cell level to that of the entire organism. GenEPi reveals transient, local mechanical stimuli in the plasma membrane of single cells, resolves repetitive contraction-triggered stimulation of beating cardiomyocytes within microtissues, and allows for robust and reliable monitoring of Piezo1-dependent activity in vivo. GenEPi will enable non-invasive optical monitoring of Piezo1 activity in mechanochemical feedback loops during development, homeostatic regulation, and disease.

Throughout an organism's lifetime, cell mechanosensation (i.e., the ability to perceive and respond to mechanical stimuli in the form of shear stress, tension, or compression) is essential in a myriad of developmental, physiological, and pathophysiological processes, including embryogenesis, homeostasis, metastasis, and wound healing[1]. How these processes incorporate active feedback via force sensing at the cellular level is an area of active study, and a wide range of tools have been developed to interrogate cell mechanics[2,3].

Stretch-activated ion channels, including the Piezo proteins, can respond to various external mechanical stimuli[4,5]. Most vertebrates have two Piezo genes, *Piezo1* and *Piezo2*[4], and functional homologs have been identified both in plants[6] and invertebrates[7]. While Piezo2 function is mainly restricted to the peripheral nervous system, Piezo1 is expressed in a wide range of tissues and has been shown to contribute to mechanotransduction in various organs[5]. Mutations in human Piezo1 have been implicated in diseases, such as dehydrated hereditary

[1]Department of Biosystems Science and Engineering, Eidgenössische Technische Hochschule (ETH) Zurich, Basel, Switzerland. [2]Department of Bioengineering, Imperial College London, London, UK. [3]Department of Molecular, Cellular and Developmental Biology, Yale University, New Haven, CT, USA. [4]Section of Investigative Medicine, Department of Metabolism, Digestion, and Reproduction, Imperial College London, London, UK. [5]Leeds Institute of Cardiovascular and Metabolic Medicine, LIGHT Laboratories, University of Leeds, Leeds, UK. [6]Molecular and Clinical Sciences Research Institute, St. George's, University of London, London, UK. [7]Department of Biochemistry, Cell and Systems Biology, Institute of Systems, Molecular and Integrative Biology, University of Liverpool, Liverpool, UK. [8]These authors contributed equally: Sine Yaganoglu, Konstantinos Kalyviotis. ✉e-mail: p.pantazis@imperial.ac.uk

stomatocytosis[8,9] and general lymphatic dysplasia[10,11], and global knockout of Piezo1 in mice causes embryonic lethality[12,13], highlighting the importance of this channel for development and homeostasis[5]. However, how cells and tissues integrate Piezo1 activity has been mainly examined by indirect outputs, such as morphological changes, protein expression, electrophysiological signaling, cytosolic calcium ($Ca^{2+}$) imaging, and transcriptional activity in response to mechanical stimuli[14].

Here, we rationally engineer and validate a genetically-encoded fluorescent reporter of Piezo1-dependent activity, named GenEPi, which has broad applicability across biological scales. We demonstrate that GenEPi has a high spatiotemporal resolution to report Piezo1-dependent activity from the single-cell level to that of the entire organism. We use Total Internal Reflection Fluorescence Microscopy (TIRFM) to showcase that GenEPi reveals transient, local mechanical stimuli in the plasma membrane of single cells. We demonstrate that GenEPi resolves repetitive contraction-triggered mechanical stimulation of beating cardiomyocytes within cardiac microtissues derived from mouse embryonic stem cells (ESCs). Finally, we validate GenEPi in vivo by generating a GenEPi zebrafish transgenic line which allows for robust and reliable monitoring of Piezo1-dependent activity using non-invasive optical imaging. Our work establishes GenEPi as a versatile and powerful tool for studying mechanosensitive processes, providing invaluable insights into Piezo1 dynamics in diverse biological contexts.

## Results

### Reporter engineering

In order to develop a non-invasive, genetically-encoded, mechanosensitive fluorescent reporter that is applicable to a wide variety of cells and types of mechanical stimuli, we set out to design a reporter of

Piezo1 activity. It has been shown that the C-terminus of Piezo1 resides in the cytosol and contains the ion-permeating channel[15,16], which has a preference for divalent cations, such as $Ca^{2+}$ [15,16]. Upon opening, $Ca^{2+}$ concentration near the channel, referred to as the $Ca^{2+}$ microdomain, is typically several-fold higher than resting levels[17]. We, therefore, hypothesized that by targeting a Genetically-Encoded $Ca^{2+}$ Indicator (GECI) to the ion-permeating channel of Piezo1, we can obtain an optical readout for its activity.

We reasoned that a fluorescent reporter of channel activation would require a GECI with low $Ca^{2+}$ affinity and a wide dynamic range to reliably monitor the considerable $Ca^{2+}$ increase in the microdomains, while displaying a low response to cytosolic $Ca^{2+}$, which serves as an important secondary messenger in many other cellular processes[17]. To meet these requirements, we decided to evaluate GCaMPs, a class of GECIs[18], as fluorescent reporters of the Piezo1 function. In contrast to FRET-based GECIs, GCaMPs occupy a narrower spectral range, allowing for the simultaneous imaging of multiple fluorescent markers. Progressive protein engineering efforts have yielded GCaMP variants that display a wide dynamic range of response with high signal-to-noise ratios (SNR)[19].

In a systematic screen, we generated a library of reporters by fusing five different low-affinity GCaMPs[20,21] (here denoted as GCaMP-G1 to GCaMP-G5) (with $K_d$-s in the 0.6 to 6 μM range) to the C-terminus of human Piezo1 (Fig. 1a). Given the influence of linker length on the sensing mechanism[22], we employed flexible linker peptides with varying lengths to attach GCaMPs to Piezo1 (Fig. 1b). The generated variants were evaluated based on their response to both mechanical stimuli and cytosolic $Ca^{2+}$ fluctuations that were independent of Piezo1 activity. To test their responses to mechanical stimuli, variants were exposed to physiological levels of fluid shear stress[23] (see Methods) (Fig. 1c), which causes a Piezo1-dependent $Ca^{2+}$ increase in HEK293T cells[12].

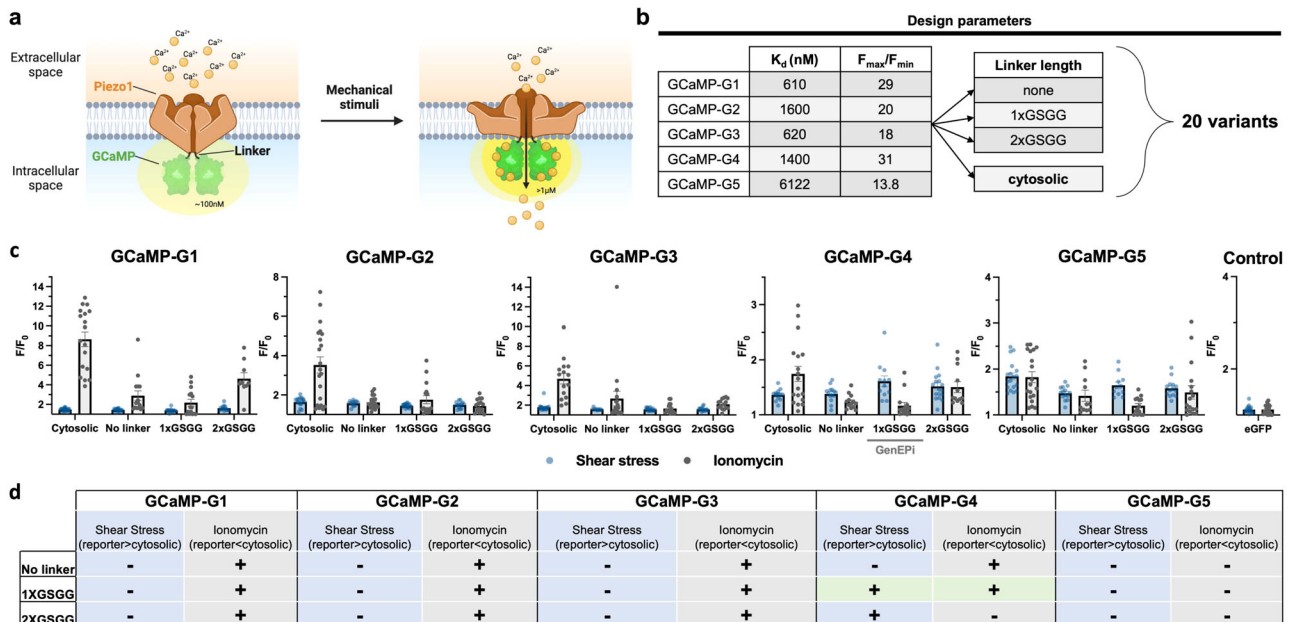

**Fig. 1 | Design and systematic screen to identify an optical reporter of Piezo1 activity. a** Schematic of the reporter working principle. A low-affinity GCaMP is targeted near the C-terminal site of the Piezo1 channel. When mechanical stimuli induce channel opening, incoming $Ca^{2+}$ (in yellow) binds to GCaMP, causing an increase in green fluorescence. **b** Design of the reporter screen. Five different GCaMPs with low affinity for $Ca^{2+}$ (GCaMP-G1 to -G5) were fused to the C-terminus of human Piezo1 without any linker, with the addition of a short linker (i.e., 1xGly-Ser-Gly-Gly (1xGSGG) linker sequence) and long linker (i.e., 2xGSGG linker sequence). **c** Results of the reporter screen. The response of the resulting 20 variants (including the co-transfected cytosolic GCaMPs and human Piezo1) to shear stress (10 dyne/cm², blue) and ionomycin (1 μM, grey) and identified Piezo1-

1xGSGG-GCaMP-G4 (GenEPi) as a specific reporter for mechanical stimuli. eGFP fused to human Piezo1 serves as a control for the noise acquired during imaging under either stimulus. Data are presented as means ± SEM. Data from three independent experiments. For detailed statistical information for the screen, see Supplementary Table 3. **d** Summary of systematic screening results that illustrate which tested variant fulfilled our key requirements (here: higher response to shear stress and lower response to ionomycin administration compared to the cytosolic response). The only variant which fulfilled both initial requirements (statistically significant) was the fusion Piezo1-1XGSGG-GCaMP-G4 which is labeled with green shading. Source data are provided as a Source Data file.

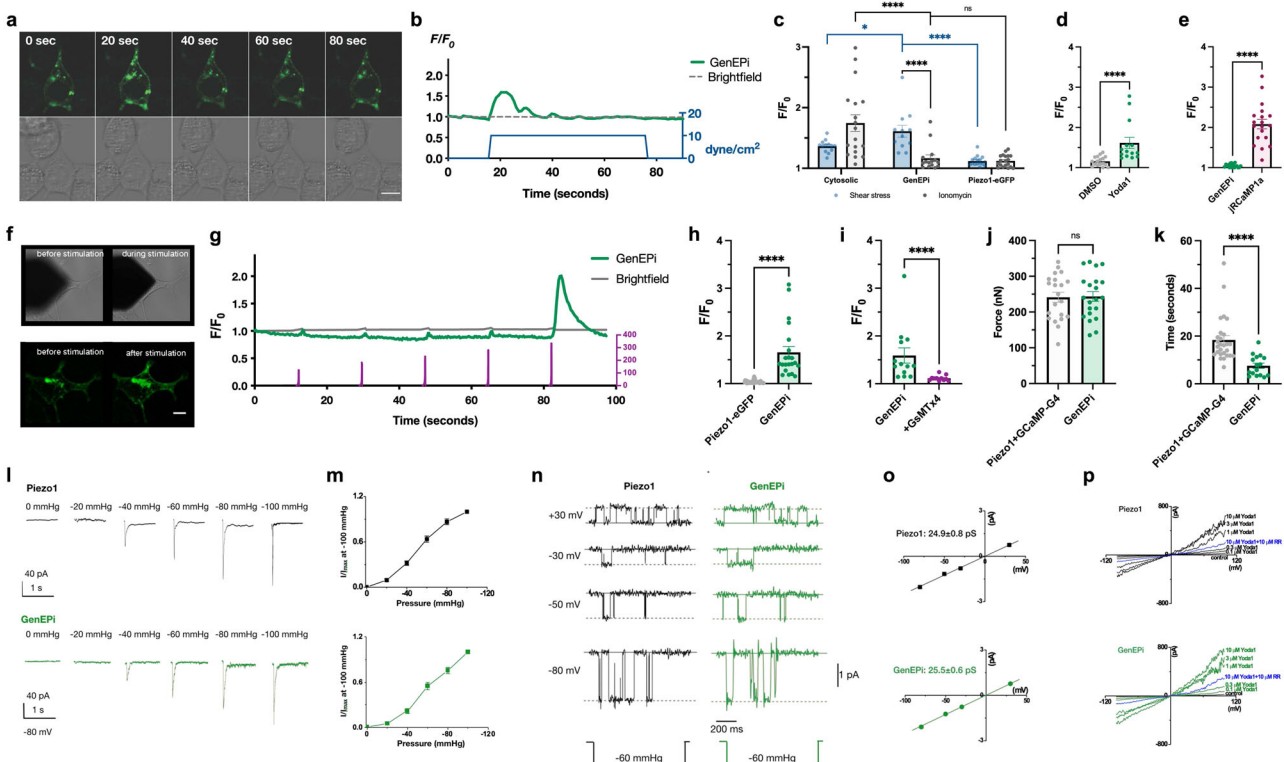

**Fig. 2 | GenEPi specifically reports Piezo1-dependent activity without affecting the functionality of the channel. a** Representative example of GenEPi activation and **b** $F/F_0$ signal intensity profile (black) in response to 10 dyne/cm$^2$ shear stress (green) in HEK293T cells. Time stamps in the images correspond to the stimulation and response profile in the graph. Scale bar, 10 µm. **c** Response of HEK293T cells expressing Piezo1 and GCaMP-G4 ($n = 13$, shear stress; $n = 18$, ionomycin), GenEPi ($n = 12$, shear stress; $n = 15$, ionomycin), or Piezo1-eGFP ($n = 16$, shear stress, $n = 19$, ionomycin) to shear stress and ionomycin. Two-tailed Mann–Whitney test, ****$p < 0.0001$; *$p < 0.05$; n.s. = $p > 0.05$, data from three independent experiments. Data were presented as means ± SEM. **d** Response of GenEPi-expressing HEK293T cells to 10 µM Yoda1 ($n = 14$) or DMSO ($n = 17$). Two-tailed Mann–Whitney test, ****$p < 0.0001$, data from three independent experiments. Data were presented as means ± SEM. **e** Response of GenEPi and jRCaMP1a expressing HEK293T cells ($n = 19$) to intracellular Ca$^{2+}$ triggered by 30 µM ATP. Two-tailed unpaired *t*-test, ****$p < 0.0001$, data from six independent experiments. Data were presented as means ± SEM. **f** Representative images of AFM cantilever stimulation of GenEPi-expressing HEK293T cells stimulated by the compressing AFM cantilever. Brightfield image of cantilever position before (top left) and during stimulation (top right) and corresponding fluorescent images of the stimulated cell before and after stimulation. Scale bar, 10 µm. **g** The mechanical stimulation procedure of compressive forces ranging from 100−400 nN (purple) along with the brightfield (gray) and fluorescent (green) traces from the cell depicted in **f. h** Amplitude of Ca$^{2+}$ responses from GenEPi ($n = 21$), and Piezo-eGFP ($n = 45$) expressing cells. Data were presented as means ± SEM. Two-tailed Mann–Whitney test, ****$p < 0.0001$. **i** Amplitude of Ca$^{2+}$ responses of GenEPi in cells transfected with GenEPi before ($n = 13$) and after addition of 3 µM GsMTx-4 ($n = 11$). Data were presented as

means ± SEM. Two-tailed Mann–Whitney test, ****$p < 0.0001$. **j** Threshold forces and pressures for cells co-transfected with human Piezo1 and cytosolic GCaMP-G4 ($n = 21$), n.s. = $p > 0.05$, Data were presented as means ± SEM. Two-tailed unpaired *t*-test. **k** Duration of Ca$^{2+}$ responses from cells co-transfected with cytosolic GCaMP-G4 and human Piezo1 ($n = 27$) and GenEPi ($n = 16$). Two-tailed Mann–Whitney test, ****$p < 0.0001$. Data were presented as means ± SEM. Data from three independent experiments. **l** Representative traces of the currents from wild-type Piezo1 (black) and GenEPi (green) evoked by the negative pressure of 0, −20, −40, −60, −80, and −100 mmHg at −80 mV with the cell-attached recording configuration. The current evoked by individual negative pressure was normalized to the maximum evoked by the negative pressure of −100 mmHg. **m** Both wild-type Piezo1 ($n = 8$ cells; 8 experiments) (black) and GenEPi ($n = 6$ cells; 6 experiments) (green) show a similar pressure-dependent response. Data were presented as means ± SEM. Two-tailed unpaired *t*-test, $p = 0.11$, $p = 0.06$, $p = 0.20$, and $p = 0.07$ for pressure sensitivities to −20, −40, −60, and −80 mmHg between Piezo1 and GenEPi. **n** Representative traces of wild-type Piezo1 (black) or GenEPi (green) currents evoked by −60 mmHg pressure at different voltages of +30, −30, −50, and −80 mV. **o** The conductance value for wild-type Piezo1 (black) is 24.9 ± 0.8 pS ($n = 6$ cells from six experiments, mean ± SEM) and the conductance value for GenEPi (green) is 25.5 ± 0.6 pS ($n = 6$ cells from six experiments, mean ± SEM). Both channels show the typical 25 pS conductance of the Piezo1 channel, indicating that the ion selectivity of Piezo1 is preserved within GenEPi. **p** Example traces of wild-type Piezo1 channels (black) or GenEPi (green) activated by 0.1, 0.3, 1, 3, 10 µM Yoda1 in the absence of mechanical stimulation. After the channel activities were evoked by 10 µM Yoda1, 10 µM ruthenium red (RR) was used to inhibit the currents (blue). Source data are provided as a Source Data file.

To determine the sensitivity of the variants to intracellular Ca$^{2+}$ levels independent of Piezo1 function, we recorded their response to the Ca$^{2+}$ ionophore ionomycin (Fig. 1c)[24]. Among the candidates tested, we identified one Piezo1-GCaMP fusion variant that satisfied our requirements (here: robust response to shear stress) (Fig. 1d), Piezo1-1xGSGG-GCaMP-G4 (containing the GCaMP6s RS1 EF4 variant[21], hereby referred to as GenEPi (Fig. 1c).

### Reporter characterization

The systemic expression of GenEPi required to confer mechanical sensitivity in HEK293T cells[25] (Supplementary Fig. 1) did not affect their

viability (Supplementary Fig. 2). The localization of GenEPi in plasma membrane and endoplasmic reticulum (ER) reflected that of wild-type Piezo1 while loss-of-function GenEPi-S217L mutant showed ER-retention and affected functionality as previously described in ref. 26 (Supplementary Note 1 and Supplementary Fig. 3).

The optical response of GenEPi (Supplementary Note 1) to fluid shear stress (Fig. 2a–c) was considerably higher (1.61 ± 0.09, mean ± SEM, $n = 12$ cells) than that of cytosolic GCaMP-G4 (1.36 ± 0.02, $n = 13$ cells) (Fig. 2c), indicating that channel tethering of GCaMP-G4 in this particular configuration provides optimal access to high Ca$^{2+}$ levels upon Piezo1 channel opening. As GenEPi retained the low affinity for

$Ca^{2+}$ (Supplementary Fig. 4), it had a low level of response to cytosolic $Ca^{2+}$ induced by ionomycin (1.16 ± 0.05, $n = 15$ cells), indistinguishable from the response levels of the control fusion protein, Piezo1-eGFP (1.12 ± 0.02, $n = 19$ cells) (Fig. 2c). In contrast, cytosolic GCaMP-G4 could not distinguish between shear stress and ionomycin and responded to both stimuli (Fig. 2c). Changing the level (1–30 dyne/cm$^2$) or duration (10–120 s) of fluid shear stress did not result in any significant difference in GenEPi response (Supplementary Fig. 5), which confirms the highly cooperative nature of GCaMP-G4 to $Ca^{2+}$ binding[21]. Notably, the tethering of all investigated GCaMP variants to the Piezo1 channel consistently reduced their response to cytosolic $Ca^{2+}$ evoked by ionomycin (Fig. 1c) which suggests that genetically-encoded $Ca^{2+}$ indicators placed near the channel are protected from cytosolic $Ca^{2+}$ fluctuations, supporting the $Ca^{2+}$ microdomain hypothesis[17].

After we confirmed the functional specificity of GenEPi to the Piezo1-specific agonist Yoda1[27] (Fig. 2d and Supplementary Fig. 6) and the dependence of its response to extracellular calcium (Supplementary Fig. 7), we determined GenEPi response to physiological $Ca^{2+}$ signaling in the cell upon addition of ATP (see Methods). We detected an ATP-dependent cytosolic $Ca^{2+}$ increase using the $Ca^{2+}$ indicator jRCaMP1a[28] (Fig. 2e and Supplementary Fig. 8), yet these elevated $Ca^{2+}$ levels were not detected by GenEPi (Supplementary Fig. 8). These results indicate that GenEPi is indeed responding specifically to Piezo1-dependent activity and does not sense physiological fluctuations of cytosolic $Ca^{2+}$, whereas cytosolic $Ca^{2+}$ indicators respond to both Piezo1-dependent and Piezo1-independent stimuli (Fig. 2c, 2e and Supplementary Fig. 8). The specificity of GenEPi response was further corroborated by the observation that membrane localization of GCaMP-G4 was not sufficient to confer functional specificity (Supplementary Note 2).

As Piezo1 is known to respond to other forms of mechanical stimuli, such as compression, we characterized the force sensitivity and temporal kinetics of GenEPi under this stimulus. We turned to a previously described Atomic Force Microscopy (AFM)-based setup[29] that allows probing Piezo1 sensitivity to mechanical stimuli while simultaneously recording the optical response of GenEPi (Fig. 2f). We applied precisely timed compressive forces ranging from 100 to 400 nN in 50 nN increments on single HEK293T cells expressing GenEPi using a 5 μm bead attached to an AFM cantilever (Fig. 2g). These compressive forces are related to pressures ranging from 2.6 to 10.2 kPa or 19.1 to 76.5 mmHg (Supplementary Note 3). GenEPi responded to short (250 ms) compressive forces with fast kinetics, but on average with comparable signal amplitude to shear stress (1.65 ± 0.12, $n = 21$ cells) (Fig. 2h). GenEPi signals in response to compressive forces were abolished in the presence of the calcium channel inhibitor GsMTx-4[30] (Fig. 2i). In contrast, the Piezo1-eGFP fusion did not show any optical response (Fig. 2h).

To characterize the force sensitivity and duration of Piezo1-induced fluorescent signals reported by GenEPi and cytosolic GCaMP-G4, we applied timed compression onto GenEPi-transfected cells and control cells co-transfected with both human Piezo1 and cytosolic GCaMP-G4. Measured threshold forces were comparable for GenEPi and cytosolic GcaMP4 (243.50 ± 13.68 nN and 241.20 ± 13.87 nN, each $n = 21$ cells, respectively) (Fig. 2j), demonstrating that the mechanical sensitivity of the channel is not affected by the protein fusion. Notably, GenEPi response to the cantilever-triggered compression lasted on average 7.56 ± 1.09 s ($n = 16$ cells), which was much shorter than that of the cytosolic indicator (18.39 ± 1.84 s, $n = 27$ cells) (Fig. 2k).

To confirm that Piezo1 within GenEPi remains functional, we conducted a series of patch-clamp electrophysiology experiments. The pressure-dependent responses evoked by negative pressure applied in a cell-attached recording configuration were similar for both GenEPi and wild-type Piezo1 (Fig. 2l–m). Single channel measurements in cell-attached patches (Fig. 2n-o) showed that GenEPi has similar

conductance values as wild-type Piezo1 (GenEPi, 25.5 ± 0.6 pS and Piezo1, 24.9 ± 0.8 pS), confirming that the conductivity of Piezo1 is not affected. Likewise, the channel kinetics of Piezo1 within GenEPi were preserved in response to repetitive mechanical stimulation (Supplementary Fig. 9), the agonist Yoda1 and the generic ion channel inhibitor Ruthenium Red (RR) (Fig. 2p and Supplementary Fig. 10). Furthermore, the electrochemical inactivation kinetic of GenEPi in response to mechanical stimuli was comparable to wild-type Piezo1 and shorter than that of the Piezo1 delayed inactivation mutant R2456H[31] (Supplementary Fig. 11).

In conclusion, GenEPi provides high spatiotemporal resolution and functional specificity of Piezo1-dependent activity compared to cytosolic $Ca^{2+}$ indicators without compromising the nature and functionality of the channel.

## Analysis of transient, local mechanical stimuli on the membrane of cells

After the characterization under controlled experimental conditions (Fig. 2), we tested whether GenEPi can visualize Piezo1-dependent activity during autonomous cellular processes, such as local cellular traction forces, using total internal reflection fluorescence microscopy (TIRFM)[32–34].

First, we tested the ability of GenEPi to resolve the localization dynamics of Piezo1 on the cell membrane of different cell types that transiently expressed GenEPi using the dox-inducible XLGenEPi plasmid (Supplementary Fig. 12) (see Methods, Molecular cloning, cell culture, and transfections). After doxycycline induction to temporally control the expression of the sensor, GenEPi was evenly distributed over the ventral surface of HEK293T, HFF, and HeLa cells (Fig. 3a), reflecting the previously reported wild-type Piezo1 membrane localization in cells[32–34] (Supplementary Fig. 3). The number of GenEPi clusters per cell (Fig. 3b) was similar among the different cell types. Likewise, the area (Fig. 3c) and perimeter (Fig. 3e) of individual GenEPi clusters and their size distributions (Fig. 3d, f) did not differ between the cell lines, demonstrating that transient expression of GenEPi provides consistent localization dynamics in different cellular contexts.

As wild-type Piezo1 has been previously shown to readily diffuse in the cell membrane[32–34], we tested whether GenEPi shows similar dynamics. Time-lapse TIRFM imaging revealed that individual GenEPi clusters were mobile in the plasma membrane (Fig. 3g–i and Supplementary Movie 1). To obtain apparent diffusion coefficients, we tracked GenEPi in the plasma membrane to build trajectories of individual clusters (Fig. 3g) and plotted the Mean-Squared Displacement (MSD) of 5332 GenEPi tracks (Fig. 3h). The slope of the MSD yielded an apparent two-dimensional (2D) diffusion coefficient of -0.005 μm$^2$/s, which is in the range of previously reported values for wild-type Piezo1[32–34].

Piezo1 function has been reported to be related to autonomous cell processes in the absence of externally applied mechanical forces, such as local cellular traction forces[32]. Indeed, we observed distinguishable GenEPi responses over the ventral surface of the cells (Supplementary Fig. 13). Notably, some of the GenEPi responses lasted for several seconds (Fig. 3i and trace i2 in 3j, k). These responses are comparable in duration to GenEPi activation in response to local mechanical forces applied by an AFM cantilever (Fig. 2k) and could be the source of previously reported calcium flickers[32,35]. Taken together, GenEPi robustly reports diffusion dynamics comparable to wild-type Piezo1 channels and captures Piezo1-dependent activity to transient, local mechanical stimuli on the membrane of the cells.

## Analysis of contraction-triggered mechanical stimulation in microtissues

In order to test the functional specificity and performance of GenEPi in an in vitro three-dimensional (3D), multicellular environment, we tested its response to homeostatic cell motions, such as cardiomyocyte

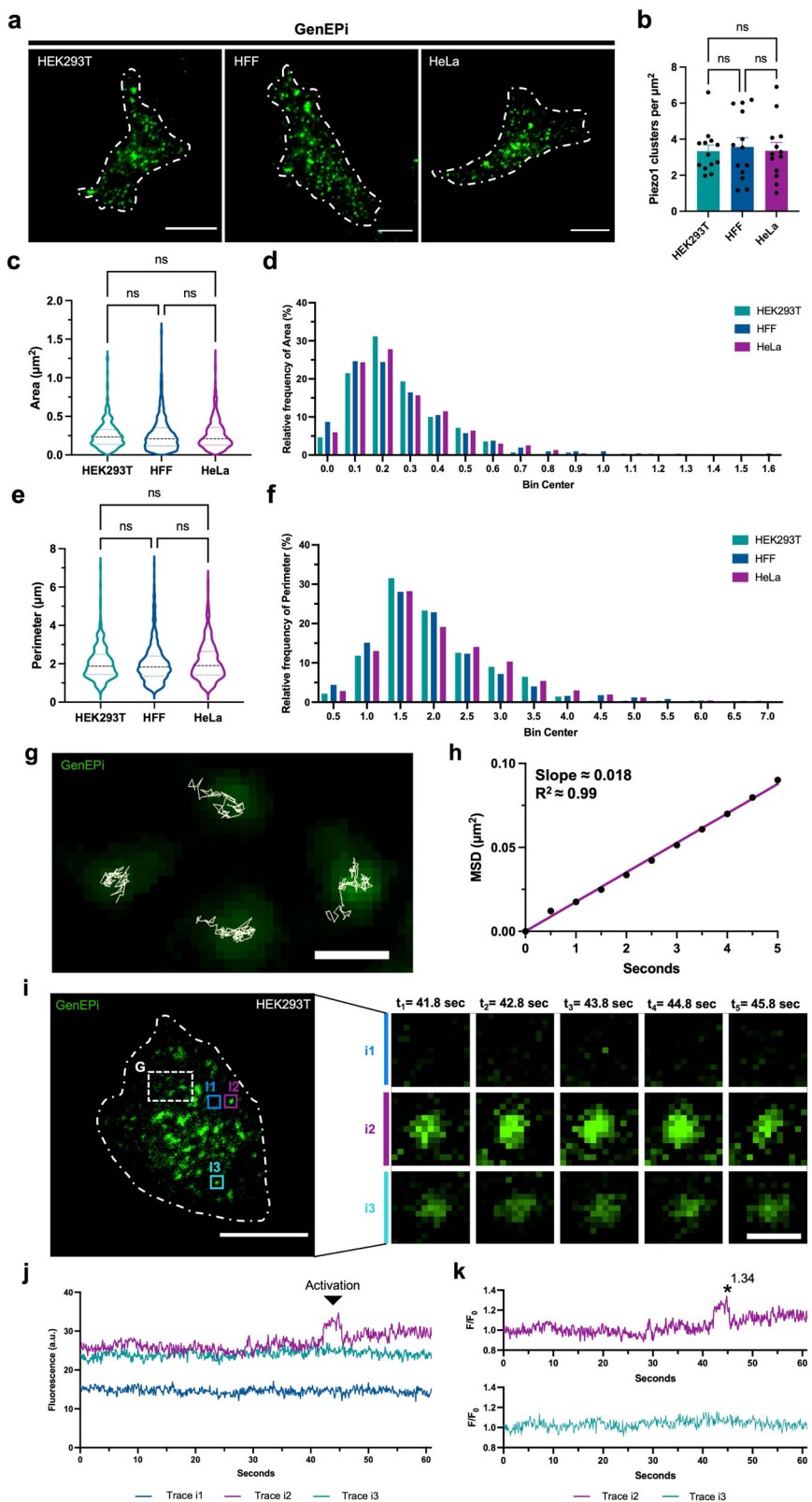

contraction where Piezo1 has been shown to act as a key mechanotransducer[36]. To this end, we generated doxycycline-inducible GenEPi mouse embryonic stem cells (mESCs)[37] (Supplementary Fig. 14) and differentiated these cells into cardiomyocytes[38]. We confirmed GenEPi activity in undifferentiated mESC by monitoring its specific response to Yoda1 (Supplementary Fig. 14). After 10 days of differentiation (Supplementary Fig. 15), regular beating patches of cells

could be identified in microtissues (Supplementary Movie 2) consisting predominantly of cardiomyocytes and other mesodermal lineage cells, such as endothelial cells (Supplementary Fig. 15). After transient induction of GenEPi expression, we observed cells that displayed noticeable GenEPi responses due to contraction-triggered mechanical stimulation within beating patches. Differentiated microtissues with multiple beating patches caused a robust systemic activation of GenEPi

**Fig. 3 | GenEPi captures Piezo1-dependent activity to transient, local mechanical stimuli on the cell membrane. a** Representative TIRFM images from the cell-substrate interface of live HEK293T, HFF, and HeLa cells expressing GenEPi. Scale bar, 10 μm. **b** Number of GenEPi clusters per cell in HEK293T, HFF, and HeLa cells expressing GenEPi. $n = 13$ cells, error bar: SEM. Ordinary one-way ANOVA test. **c–f** Quantification of GenEPi cluster area and perimeter as violin plots in HEK293T, HFF, and HeLa cells expressing GenEPi. HEK293T cells ($n = 279$), HFF cells ($n = 503$), HeLa cells ($n = 669$); $n = 1$ represent a single GenEPi (Piezo1) cluster. Kruskal–Wallis test. **g** Tracking of GenEPi (Piezo1) clusters imaged at 10 frames per second in live HEK293T cells reveals the motility of the channel clusters. The background image shows the fluorescence of GenEPi clusters captured during a single imaging frame (rectangular region (G) in image **i**, two-pixel median filter applied in FiJi/ImageJ (see Methods, Image processing and analysis). White lines depict the tracks of these

clusters over several successive frames. Scale bar, 1 μm. **h** Mean-squared displacement calculated from 5332 GenEPi tracks plotted as a function of time. Data fit to a straight line with a slope corresponding to a two-dimensional (2D) diffusion coefficient of 0.0044 μm²/s. $R^2$ for linear fit to data is 0.9969. **i** Representative time-lapse images of GenEPi (Piezo1) cluster dynamics in a single HEK293T cell expressing GenEPi. **i1–i3** (i1) Region without GenEPi cluster adjacent to a GenEPi cluster. (i2) Region with a GenEPi cluster which shows dynamic behavior and changes in fluorescence. (i3) Region with a GenEPi cluster which does not show dynamic behavior. Scale bars, 10 and 1 μm, respectively. **j** Raw values of fluorescence intensity profiles of the ROIs i1–i3 (Trace i1, Trace i2, and Trace i3) which show different GenEPi activity dynamics. **k** Fluorescence intensity profiles ($F/F_O$) of the ROIs i1–i3 which illustrate the different GenEPi activity dynamics. Source data are provided as a Source Data file.

---

in response to the autonomous regular contraction of cardiomyocytes (Fig. 4a). Notably, in dissected microtissue, where single cells were attached to autonomously beating cardiomyocytes, GenEPi responses were restricted to places of local membrane displacement (Fig. 4b and Supplementary Fig. 16). In both conditions, the response amplitude range ($F/F_0 = 1.15$ to 2.94) was comparable to that of shear stress and compressive forces, while the sub-second responses were qualitatively coupled to the autonomous regular beating of the cardiomyocytes (Fig. 4b, Supplementary Fig. 17, and Supplementary Movie 3).

In order to confirm that the source of GenEPi responses were indeed cardiomyocyte contractions, we applied blebbistatin, a myosin inhibitor, which blocks the contractions and uncouples mechanical stimuli-induced $Ca^{2+}$ influx from $Ca^{2+}$ processes accompanying spontaneous cell contractions[39]. Fast GenEPi responses decreased when contractions stopped in response to blebbistatin (Fig. 4c), as demonstrated by the significant decrease in the amplitude and frequency of the GenEPi response (Supplementary Fig. 18), confirming cardiomyocyte contractions as the source of GenEPi signals.

To complement our analysis, we tested the ability of GenEPi to respond and resolve higher-frequency cardiomyocyte contractions. To this end, we employed norepinephrine, a ligand which activates adrenoreceptor signals and increases the contraction rate in cardiomyocytes[40]. We indeed observed a significant increase in the frequency and the amplitude of GenEPi responses in proportion to the norepinephrine regime (10 nM to 10 μM) (Fig. 4d–g), validating the ability of GenEPi to temporally resolve cardiomyocyte contractions. Additionally, to further demonstrate that other channels, such as L-type channels, do not contribute to the response of GenEPi, we treated differentiated beating cardiac microtissues with nifedipine which is known to block most L-type channels and has a strong negative ionotropic effect[41–43]. Indeed, nifedipine treatment abolished the beating of cardiomyocytes within the differentiated cardiac microtissues, suggesting that L-type channels were effectively blocked (Fig. 4h–j). Yet, the baseline fluorescence of the GenEPi-expressing cells (Fig. 4k, l) was not affected by this blockade, indicating that other calcium channels, such as L-types channels, do not affect the Piezo1-dependent functional readout of GenEPi.

Overall, GenEPi shows high spatiotemporal resolution of Piezo1-dependent activity in both dissected and non-dissected microtissues, capable of specifically sensing spontaneous as well as drug-stimulated repetitive mechanical stimuli of beating cardiomyocytes.

### Monitoring Piezo1-dependent activity in vivo

There is an increasing body of work that has revealed Piezo1's central role in mediating mechanosensation in a plethora of physiological and pathological conditions (Supplementary Table 1[44–71]). Thus, visualizing its activity in vivo will allow for understanding Piezo1's role in the interplay of mechanical and biochemical signals in development and disease[72]. To investigate the ability of GenEPi to report Piezo1-dependent activity in the context of a whole organism, we generated a zebrafish transgenic line that allows for conditional manipulation of

GenEPi expression using the zebrafish heat-shock promoter hsp70-l[73] (Fig. 5a, see Methods, Molecular cloning). We reasoned that the transparency of zebrafish embryos combined with the direct optical readout of GenEPi would enable monitoring of Piezo1-dependent activity of single cells in the tissues, organs, and organ systems within the organism. GenEPi expression was induced following heat-shock of *Tg(hsp70:GenEPi)* zebrafish for 1 h at 37.5 °C (Fig. 5b). Compared to non-heat-shocked embryos, GenEPi expression was systematic and reached a steady fluorescence state at ~4–5 h post-heat-shock (Fig. 5c–e and Supplementary Fig. 19). Like previous in vitro results (Supplementary Fig. 2), GenEPi signal was mainly localized on the plasma membrane of different cell types, such as endocardium cells, neural retina cells and keratinocytes (Fig. 5f and Supplementary Fig. 20). The temporal induction of systemic expression of GenEPi did not affect the development of embryos to adulthood and the heart rate of individual embryos did not differ in comparison to control non-heat-shocked embryos when the latter recovered from the heat-shock process (Supplementary Fig. 21).

As zebrafish are potent models for drug and toxicity screening[74,75] and only a few modulators of Piezo1 activity are known[27,76], we assessed the activity of GenEPi upon pharmacological intervention in vivo. As a proof-of-principle, we used the agonist Yoda1 and the generic ion channel inhibitor RR to probe Piezo1 function following a similar approach to our in vitro analyses and electrophysiology experiments (Fig. 2d, p). Heat-shocked *Tg(hsp70:GenEPi)* zebrafish embryos in egg medium (E3 medium) were sequentially incubated in 10 μM Yoda1, 10 μM Yoda1 and 10 μM RR, and 10 μM RR followed by washing out in E3 medium (see Methods, Zebrafish transgenesis and in vivo pharmacological intervention experiments). To enable rapid interpretation of GenEPi response over a large field of view, we used a widefield fluorescence microscope. We focused on the developing eye, a large organ in direct contact with the egg medium easily accessible for imaging and pharmacological interventions. Heat-shocked *Tg(hsp70:GenEPi)* fish showed instantaneous and significant fluorescence increase in response to 10 μM Yoda1, whereas transfer to a solution containing 10 μM Yoda1 and 10 μM of RR inhibited GenEPi activation (Fig. 5g, h), consistent with our electrophysiology results (Fig. 2p). Zebrafish transferred to 10 μM RR demonstrated further decrease in fluorescence intensity (Fig. 5g, h). Notably, washing out of the zebrafish in the original E3 medium restored the fluorescence of GenEPi to pre-treatment activity levels (Fig. 5g, h), demonstrating that GenEPi retains the pharmacological properties of Piezo1.

We previously showed (Fig. 4) that GenEPi can report cardiomyocyte contraction-triggered mechanical stimulation with high spatiotemporal resolution within 3D cardiac microtissues. To confirm that GenEPi can also monitor Piezo1-dependent activity during heart beating in vivo, we imaged heat-shocked *Tg(hsp70:GenEPi)* zebrafish embryo hearts at 3dpf (Supplementary Movie 4). GenEPi is strongly expressed in both cell layers of the developing zebrafish heart (Fig. 5i) once fluorescence reaches a steady state. Performing dynamic time-lapse widefield imaging, we observed a mechanical stimuli-dependent

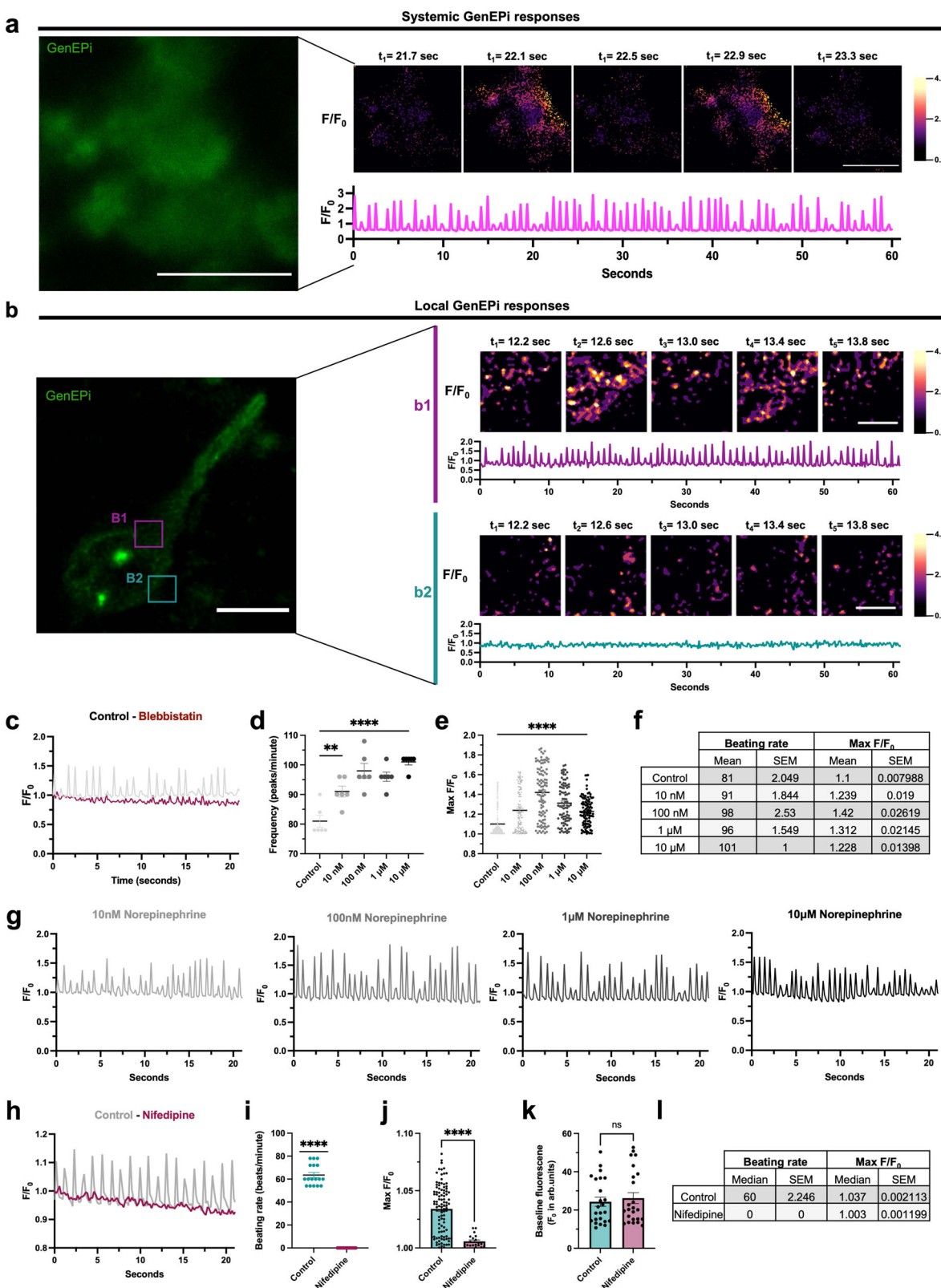

activation of GenEPi in the atrioventricular canal (AVC) of the heart (Fig. 5j–l and Supplementary Fig. 22). The GenEPi activation was abolished when the zebrafish heart was stopped with 2,3-Butanedione 2-monoxime (BDM) treatment (Fig. 5j–l), confirming the mechanical nature of GenEPi activation.

Overall, we confirm that the ability of GenEPi to resolve Piezo1-dependent activity is preserved in vivo and demonstrate a robust and reliable in vivo GenEPi system that is able to generate information-rich and physiologically relevant readouts.

## Discussion

In summary, we introduce GenEPi as a designed intensiometric, genetically-encoded reporter for Piezo1-dependent activity. GenEPi provides a highly specific and non-invasive functional readout from

**Fig. 4 | GenEPi reports cardiomyocyte contraction-triggered mechanical stimulation with high spatiotemporal resolution. a** Multicellular and systemic activation of GenEPi in response to autonomous contraction of cardiomyocytes in differentiated tissue within multiple beating patches. Representative time-lapse fluorescence intensity ($F/F_O$) images and profile of multiple responding cells embedded in the differentiated tissue of beating cardiomyocyte patches. Scale bar, 10 μm. **b** Single cell local responses of GenEPi in a dissected beating patch in response to the autonomous beating of cardiomyocytes. **b1, b2** Representative time-lapse fluorescence intensity ($F/F_O$) images and profiles of a responding (**b1**) and a non-responding (**b2**) ROI in a single cell attached to the beating cardiomyocyte patch. Scale bar, 5 μm. **c** Representative fluorescence intensity ($F/F_O$) profile from an ROI with systemic activation of GenEPi in response to cardiomyocyte contraction before (control) and after the addition of 100 μM blebbistatin. **d** Frequency of peaks, **e** amplitude of fluorescence intensity ($F/F_O$) changes and **f** median ± SEM values of **d** and **e** before (control) and after the addition of norepinephrine (range of concentrations, 10 nM–10 μM). $n = 6$ ROIs, error bar: SEM. Ordinary one-way ANOVA and Kruskal–Wallis test, respectively. Data from three independent experiments. **g** Representative fluorescence intensity ($F/F_O$) profiles from an ROI with GenEPi activation in response to cardiomyocyte contraction after the addition of norepinephrine (range of concentrations, 10 nm–10 μM). **h** Representative fluorescence intensity ($F/F_O$) profile from an ROI with systemic activation of GenEPi in response to cardiomyocyte contraction before (control) and after the addition of 100 nM nifedipine. **i** Beating rate of cardiac microtissues before and after the addition of 100 nM nifedipine which abolishes the contraction of the beating cardiomyocytes. $n = 17$ measurements from three independent biological replicates, error bar: SEM. Two-tailed Wilcoxon rank-sum test. **j** Amplitude of fluorescence intensity ($F/F_O$) changes before (control) and after the addition of 100 nM nifedipine. $n = 101$ and $n = 20$ measurements for control and nifedipine treatments, respectively, from three individual biological replicates, error bar: SEM. Two-tailed Mann–Whitney test. **k** Baseline fluorescence ($F_O$) before and after the addition of 100 nM nifedipine. $n = 24$ ROIs containing multiple cells for each condition from 3 individual biological replicates, error bars: SEM. Two-tailed Mann–Whitney test. **l** Median ± SEM values of (**i**) and (**j**) before (control) and after the addition of 100 nM nifedipine. Source data are provided as a Source Data file.

the single-cell level to that of the entire organism. The engineering of GenEPi was achieved by successfully targeting a low-affinity and high dynamic range GCaMP to the $Ca^{2+}$ microdomain of the Piezo1 channel using a flexible linker of defined length. Our systematic screening results highlight the importance of the amino acid composition and length of the flexible linker used to secure reporting of only Piezo1-dependent $Ca^{2+}$ signals. Indeed, the duplication of the flexible linker unit (GSGG) was sufficient to abolish the specificity to Piezo1-dependent stimuli (see Fig. 1c). While previous work suggests that a C-terminal GFP fusion to mouse Piezo1 without the use of a flexible linker can alter the response of the channel[77], our conductivity and electrophysiological analyses of GenEPi demonstrate that our approach does not affect the functionality of the channel and faithfully reflects the responses of wild-type Piezo1 (see Fig. 2l–p). Hence, we suggest that our design principle of GenEPi can serve as a blueprint for developing and engineering optical reporters of other ion channels without affecting their function.

Due to the highly cooperative $Ca^{2+}$ sensing mechanism of the GCaMP, GenEPi is able to robustly report $Ca^{2+}$ influx upon channel opening with high spatiotemporal resolution, yet different forces and activation states can only be comparatively correlated to the fluorescence response. Future engineering incorporating improved fluorescent probes could secure quantitative readout of applied forces. The straightforward intensiometric visualization of Piezo1-dependent activity using GenEPi unifies the previously separated readouts of channel function and channel dynamics. GenEPi can be readily implemented in a multitude of cellular systems without noticeable adverse physiological effects under temporally controlled overexpression and produces robust readouts in various stimulation conditions (see Fig. 5). In the future, targeted CRISPR/Cas approaches to tag endogenous Piezo1 will promote our understanding of how Piezo1 senses and interprets a variety of intrinsic and extrinsic mechanical and recently discovered non-mechanical stimuli[78].

Our zebrafish model demonstrates the universality of GenEPi to resolve Piezo1-dependent stimulation both in vitro and in vivo. The GenEPi transgenic line will be an essential tool to survey hypotheses of previous in vitro results and to explore Piezo1-dependent activity in various pathophysiological conditions. Considering the similarity of GenEPi responses during zebrafish heart beating to previously characterized shear stress forces in the zebrafish heart[79,80], we anticipate that GenEPi will be a powerful tool to investigate Piezo1-dependent mechanical signals in cardiac development and morphogenesis[81]. So far, only a few modulators of Piezo1 activity have been identified, highlighting the need for new targeted drug discovery pipelines. The absence of a specific functional readout of Piezo1 activity has hampered an effective discovery of Piezo1-specific drugs. Our GenEPi

zebrafish can serve as a cost-effective and scalable in vivo pharmacological intervention system. The direct imaging of compound effects on Piezo1 activity simultaneously in different organs will enable to distinguish between systemic and tissue-specific modulation of channel function. This refined approach will create the opportunity to select lead compounds with minimal side effects considering the plethora of functions Piezo1 channels exert[82].

Altogether, we introduce GenEPi as an ideal tool to elucidate the full extent to which Piezo1-dependent signaling regulates development, physiology, and disease.

## Methods
### Molecular cloning
We obtained the human Piezo1 cDNA from Kazusa Inc, Japan. Generation of the first four types of GCaMPs; mGCaMP6s-EF4 (GCaMP-G1), mGCaMP6f-EF4 (GCaMP-G2), mGCaMP6s RS1-EF3 (GCaMP-G3), and mGCaMP6s RS1-EF4 (GCaMP-G4), were described elsewhere[21]. Fast-GCaMP-EF20 (here denoted as GCaMP-G5) was a gift from Samuel Wang (Addgene plasmid #52645)[20]. pGP-CMV-NES-jRCaMP1a was a gift from Douglas Kim (Addgene plasmid #61462)[28].

Piezo1 was amplified using Herculase II fusion DNA Polymerase (600675, Agilent Technologies) and all $Ca^{2+}$ indicators with various linker lengths were amplified with Phusion high-fidelity DNA polymerase (M0530, NEB). A list of primers, ordered from Sigma-Aldrich, can be found in Supplementary Table 2. Piezo1 and the $Ca^{2+}$ indicators were introduced using restriction cloning and T4 DNA ligase (M0202, NEB). The Lck targeting sequence flanking restriction sites were synthesized by Genewiz and introduced upstream of GCaMP-G4 and GCaMP-G5. All restriction enzymes were purchased from NEB. PCR and digestion products were purified using a QIAquick PCR purification kit (28104, Qiagen) and QIAquick Gel Extraction kit (28704, Qiagen). Ligations were carried out using T4 Ligase (NEB) at 24 °C for 1 h, followed by chemical transformation using Turbo ultracompetent *E. coli* based on K12 strain (NEB) and grown on Agar LB plates (Q60120 and Q61020, Thermo Fisher) and LB liquid media (244610, BD Bioscience) supplemented with appropriate antibiotics (100 μg ml⁻¹ Ampicillin or 50 μg ml⁻¹ Kanamycin, Sigma-Aldrich). Clones were screened using restriction digest and sequenced by Microsynth. Plasmid DNA isolation was carried out using ZR Plasmid Miniprep (D4054, Zymo Research). The pCMV-GenEPi plasmid is deposited on Addgene (plasmid #140236).

The p2Lox-GenEPi plasmid was prepared by subcloning GenEPi to the p2Lox backbone using traditional restriction enzyme cloning. The original pCMV-GenEPi plasmid (Addgene plasmid #140236) and the p2Lox vector were both digested with NotI/HindIII to create compatible sticky ends and ligated using T4 Ligase (M0202, NEB). Clones were screened using restriction digest and sequenced

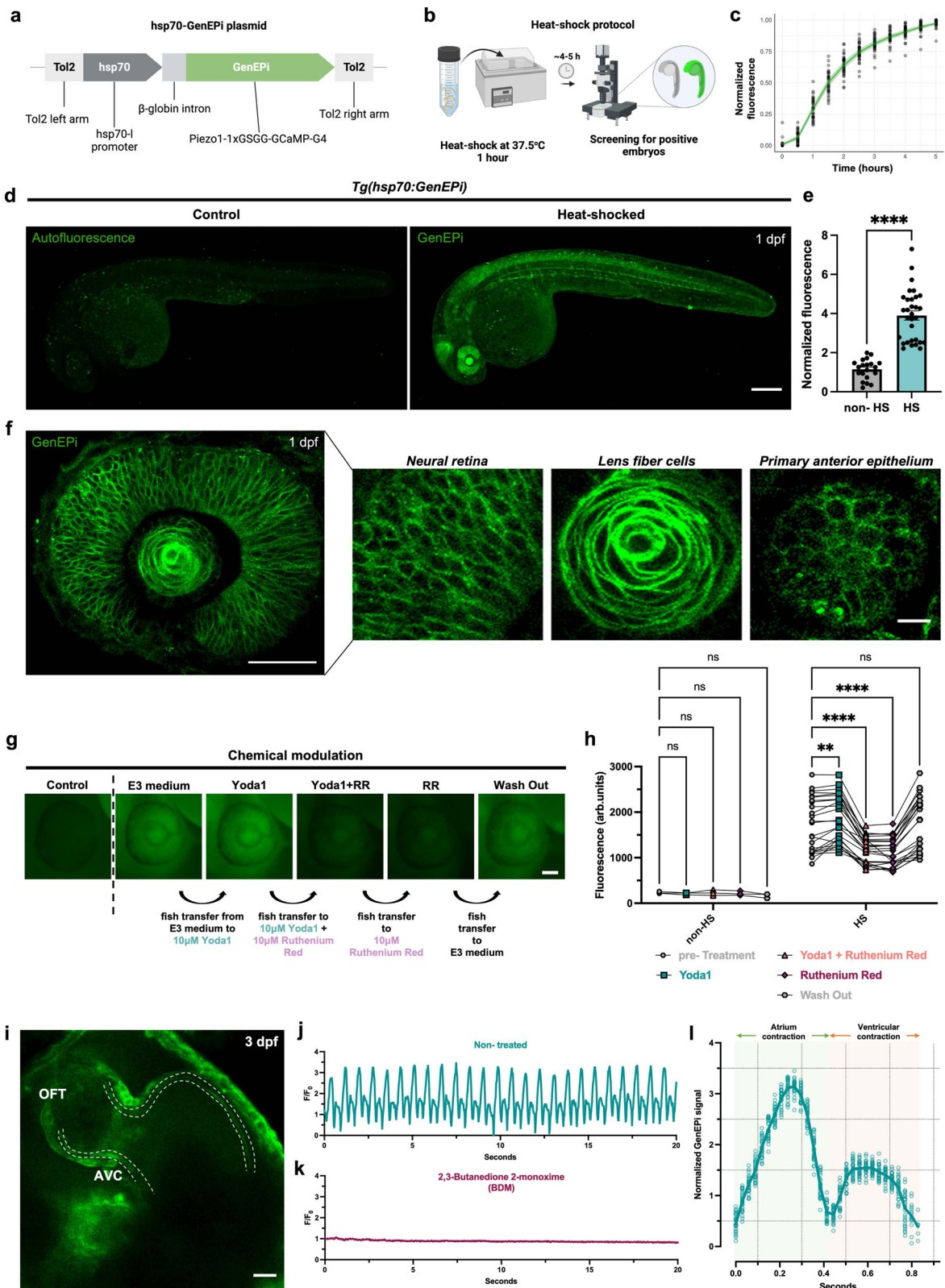

by Microsynth using custom-made sequencing primers. Plasmid DNA isolation was carried out using ZR Plasmid Miniprep (D4054, Zymo Research).

The XLone-GenEPi (XLGenEPi) plasmid was prepared by subcloning GenEPi to the XLone-EGFP backbone with traditional restriction enzyme cloning. XLone-GFP[83] was a gift from Xiaojun Lian (Addgene plasmid #96930; http://n2t.net/addgene:96930; RRID: Addgene_96930). The

original pCMV-GenEPi (Addgene #140236) and XLone-EGFP plasmids were digested with HindIII and KpnI, respectively. Both sticky ends were blunted (HindIII filled-in, KpnI overhangs removed) using the Quick Blunting Kit (NEB, E201), digested with SphI, and ligated using T4 Ligase (M0202, NEB). Clones were screened using restriction digest and sequenced by Eurofins Genomics (https://eurofinsgenomics.eu/) using our custom-made sequencing primers (see Supplementary Table 2).

**Fig. 5 | GenEPi reports Piezo1-dependent activity changes in vivo. a** Schematic of the pTol2-hsp70:GenEPi plasmid used for the zebrafish transgenesis. GenEPi is downstream of the zebrafish hsp70-l promoter after the β-globin rabbit intron. **b** Schematic of the heat-shock protocol used to induce the expression of GenEPi in *Tg(hsp70:GenEPi)* zebrafish. **c** Longitudinal analysis of GenEPi expression upon heat-shock of *Tg(hsp70:GenEPi)* zebrafish at 1dpf, fluorescence steady state is reached after ~4–5 h. *n* = 35 fish. **d** Representative fluorescence of a non- (non-HS) and heat-shocked (HS) *Tg(hsp70:GenEPi)* zebrafish at 1 day post-fertilization (dpf). Scale bar, 200 μm. **e** Increase in fluorescence intensity following the heat-shock protocol in (**b**). (*n* = 20 and *n* = 31 non-heat-shocked (non-HS) and heat-shocked (HS) fish, respectively). Data were presented as means ± SEM. **f** GenEPi expression in the developing eye, neural retina, lens fiber cells, and primary anterior epithelium in a 1dpf *Tg(hsp70:GenEPi)* zebrafish. Scale bar, 20, 200, and 20 μm, respectively. **g** Representative images of GenEPi expression in the eye of a 1dpf *Tg(hsp70:GenEPi)* zebrafish during the chemical modulation of channel's activity by Yoda1 and RR. Scale bar, 50 μm. **h** Normalized fluorescence intensity of control and *Tg(hsp70:GenEPi)* zebrafish during the chemical modulation of GenEPi activity in 10 μM Yoda1, 10 μM Yoda1 and 10 μM RR, 10 μM RR, and wash out to E3 medium. Non-heat-shocked embryos which do not express GenEPi showed non-significant responses both in the presence or absence of Yoda1 or RR (paired data of *n* = 3 and *n* = 24 non-HS and HS transgenic zebrafish, respectively). Data were presented as means ± SEM. Two-way ANOVA test. **i** Representative image of a *Tg(hsp70:GenEPi)* zebrafish heart at 3dpf. GenEPi is expressed in both the endocardium (dashed line) and myocardium cells in the developing zebrafish heart. Scale bar, 20 μm. **j** Representative fluorescence intensity ($F/F_0$) profile from the antrioventricular canal (AVC) of a *Tg(hsp70:GenEPi)* zebrafish heart which shows the two-phase GenEPi activation. **k** Representative fluorescence intensity ($F/F_0$) profile from the same ROI in **j** at the AVC of the same *Tg(hsp70:GenEPi)* zebrafish heart after treatment with BDM which abolishes heart beating, showing absence of GenEPi responses. **l** Mean of 27 representative GenEPi responses during individual heart beating cycles at 1dpf. Normalization of the GenEPi signal was obtained by computing the ratiometric intensity of GenEPi to the average of 12 NLS-mCherry signals of individual heartbeats in *Tg(kdrl:NLS-mCherry)* zebrafish at 1dpf. Source data are provided as a Source Data file.

Plasmid DNA isolation was carried out using Qiagen Plasmid Midi Kit (Qiagen, 12143).

The hsp70:GenEPi plasmid was prepared using Gibson Assembly (NEB, E2611L) following the manufacturer's instructions. The backbone vector containing the 1517 bp hsp70-l promoter sequence was amplified with PCR using the Gibson Assembly primers (see Supplementary Table 2). The original pCMV-GenEPi plasmid (Addgene plasmid #140236) was linearized using NotI/HindIII. Clones were screened using restriction digest and sequenced by Eurofins Genomics (https://eurofinsgenomics.eu/) using our custom-made sequencing primers (see Supplementary Table 2). Plasmid DNA isolation was carried out using Qiagen Plasmid Midi Kit (Qiagen, 12143).

For site-directed mutagenesis, the Q5 Site-Directed Mutagenesis Kit (NEB, E0554S) was used following the manufacturer's instructions. The site-directed mutagenesis primers were designed using NEBaseChanger (https://nebasechanger.neb.com/) and they are listed in Supplementary Table 2. Clones were sequenced by Eurofins Genomics (https://eurofinsgenomics.eu/). Plasmid DNA isolation was carried out using Qiagen Plasmid Midi Kit (Qiagen, 12143).

## Cell culture and transfection

HEK293T, HFF, and HeLa cells were cultured at 37 °C, 5% CO₂, in high glucose DMEM with GlutaMAX (10569010, Thermo Fisher), supplemented with 10% FBS (P40-37500, Pan Biotech) and 1X Penicillin-Streptomycin solution (15140122, Thermo Fisher). Cells were routinely tested and were negative for mycoplasma infection using a Mycoplasma detection kit (B39032, LuBioScience GmBH). Plasmid DNA for transfections was isolated from 50 ml LB culture (244610, BD Bioscience) containing appropriate antibiotics using the Zymopure Plasmid Midiprep kit (D4200, Zymo Research) or the Qiagen Plasmid Midi Kit (Qiagen, 12143). The amount of DNA was measured using the Nanodrop 2000c or NanoDrop One$_c$ Spectrophotometer (Thermo Fisher).

For fluid shear stress experiments and to test the response of the reporter to various chemicals, 400–800 ng of each plasmid was introduced into cells using nucleofection (V4XC-2024, Lonza) following the manufacturer's instructions. At 24 h post-transfection, the cells were dissociated and seeded onto ibitreat flow chambers (Ibidi u-slide-VI 0.4, 80606, Ibidi GmbH) or ibitreat coated eight-well slides (80826, Ibidi GmbH) with a density of 75,000 cells per channel or well. We used 1 μM ionomycin (I3909-1ML, Sigma-Aldrich), 30 μM ATP (A6559-25UMO, Sigma-Aldrich) diluted in DPBS, 10 μM Yoda1 (5586, Tocris Bioscience) diluted in DMSO (D8418, Sigma-Aldrich), or 2.5 μM GsMTx-4 (Pepta Nova GmBH) diluted in water.

For AFM experiments, HEK293T cells were transfected using lipofection (11668019, Lipofectamine 2000, Thermo Fisher) following the manufacturer's instructions. At 24 h post-transfection, the cells were dissociated and seeded onto 35-mm-wide cover-glass-bottom Fluorodishes (FD35-100, World Precision Instruments), with a density of 300,000 cells per plate.

For the in situ affinity measurements, HeLa cells were plated on 35 mm glass-bottom culture dishes (MatTek) and allowed 24 h to adhere before transfection with FuGENE HD (Promega) following the manufacturer's instructions. Cells were maintained for 12–24 h before being used in experiments.

For TIRFM imaging experiments, HEK293T, HFF, and HeLa cells were transfected with the XLGenEPi plasmid (Tet-on, dox-inducible expression) using lipofection (L3000001, Lipofectamine 3000, Thermo Fisher) following manufacturer's instructions. At 24 h post-transfection, the cells were dissociated and seeded onto 0.1% gelatin-coated 20-mm-wide cover-glass-bottom confocal dishes (734-2906, VWR), with a density of 200,000 cells per well. About 200 ng/ml doxycycline (D9891, Merck) was added to induce GenEPi's expression 24 h prior to imaging. About 200 ng/ml doxycycline was used to induce GenEPi's expression and generate sufficient contrast, which is in the lower range of previously used concentrations[83].

For the Yoda1 and EGTA experiments, HeLa cells were transfected with the XLGenEPi plasmid (Tet-on, dox-inducible expression) using lipofection (L3000001, Lipofectamine 3000, Thermo Fisher) following manufacturer's instructions. At 24 h post-transfection, the cells were dissociated and seeded onto 0.1% gelatin-coated 20-mm-wide cover-glass-bottom confocal dishes (734-2906, VWR), with a density of 200,000 cells per well. About 200 ng/ml doxycycline (D9891, Merck) was added to induce GenEPi's expression 24 h prior to imaging. About 200 ng/ml doxycycline was used to induce GenEPi's expression and generate sufficient contrast, which is in the lower range of previously used concentrations[83]. We used several concentrations of Yoda1 (5586, Tocris Bioscience) diluted in DMSO (D8418, Sigma-Aldrich) ranging from 5 to 50 μM and 2 mM EGTA (OmniPur EGTA, 4100) diluted in water.

For the loss-of-function GenEPi experiments, HeLa cells were transfected with the XLGenEPi-S217L plasmid (Tet-on, dox-inducible expression) using lipofection (L3000001, Lipofectamine 3000, Thermo Fisher) following manufacturer's instructions. At 24 h post-transfection, the cells were dissociated and seeded onto 0.1% gelatin-coated 20-mm-wide cover-glass-bottom confocal dishes (734-2906, VWR), with a density of 200,000 cells per well. About 200 ng/ml doxycycline (D9891, Merck) was added to induce GenEPi's expression 24 h prior to imaging. We used 10 μM Yoda1 (5586, Tocris Bioscience) diluted in DMSO (D8418, Sigma-Aldrich).

## Generation of inducible GenEPi-mESC cell lines

Doxycycline-inducible GenEPi mESCs were generated using ZX1 mESCs carrying rtTA in the Rosa26 locus and dox-inducible cre flanked

by self-incompatible LoxP sites in the HPRT locus[37], kindly provided by Dr. Michael Kyba. ZX1 mESCs were cultured in DMEM (Life Technologies), 15% FBS (PAN Biotech), 2 mM L-Glutamine (Invitrogen), 1X non-essential amino acids, 0.1 mM β-mercaptoethanol, 100 U/ml leukemia inhibitory factor (Peprotech), 1 µM PD0325901 (Selleckchem), and 3 µM CHIR99201 (R&D Systems) on 0.1% gelatin-coated plates. Prior to electroporation, ZX1 mESCs were exposed to 500 ng/ml doxycycline (D9891, Merck) for 24 h. About $1 \times 10^6$ ZX1 mESCs were electroporated with 3 µg p2lox-GenEPi plasmid in a 0.4 cm electroporation cuvette at 230 mV, 500 µF and maximum resistance in a Biorad electroporator (Biorad Genepulser Xcell). Twenty-four hours after electroporation, the antibiotic selection was started with 300 µg/mL G418 (Sigma). Colonies that incorporated GenEPi were verified by FACS analysis and expanded. Dox-inducible GenEPi mESCs were differentiated into cardiomyocytes as previously described in ref. 38. GenEPi mESCs were seeded as 500 cell/20 µl in hanging drops on non-adherent plates to generate embryoid bodies (EBs) in EB medium, IMDM (Life Technologies), 20% FBS (PAN Biotech), 2 mM L-Glutamine (Invitrogen), 1X non-essential amino acids and 0.1 mM β-mercaptoethanol. After 2 days, EBs were transferred to uncoated petri dishes. From days 3–5, 1 µM XAV939 was added to the culture conditions and EBs were plated on 0.1% gelatin-coated dishes from day 4. Beating EBs appeared on day 10 of differentiation. Beating EBs were either directly imaged within the multi-patch beating microtissue environment or manually dissected and dissociated using 2 mg/ml Collagenase/Dispase (Sigma) to generate smaller beating patches and single cells (Supplementary Fig. 12). For blebbistatin experiments, 40–120 µM Blebbistatin (B0560, Sigma-Aldrich) diluted in DMSO (D8418, Sigma-Aldrich) was applied to the cells until contractions were stopped. For norepinephrine experiments, 10 nM–10 µM of Norepinephrine (1468501, Merck) diluted in PBS was applied to the cells. Under these experimental conditions, we did not observe cardiomyocyte beating irregularities such as arrhythmia.

## Determination of cell viability and cell toxicity

Cell viability was determined using trypan blue exclusion assay. Briefly, cells in triplicates seeded in 6-well tissue culture plates (Thermo Fisher) were transfected with varying concentrations of GenEPi or GCaMP-G4 and human Piezo1. At 24- and 48-h post-transfection, cells were washed with 1X PBS twice and detached using 0.05% Trypsin-EDTA (25300054, Thermo Fisher). About 10 µl of cell suspension was then mixed with 10 µl 0.4% Trypan Blue, and 10 µl of this mixture was added to the cell counting slide (C10228, Thermo Fisher) and measured using Countess II Automated cell counter (Thermo Fisher). The viability was expressed as a fold difference of the untreated samples for each time point.

To determine cell toxicity, lactate dehydrogenase (LDH) assay (Life Technologies) was used on GenEPi or GCaMP-G4 and human Piezo1 transfected cells according to the manufacturer's instructions. Briefly, GenEPi or GCaMP-G4 and human Piezo1 transfected cells in triplicates were seeded in 96-well tissue culture plates (167008, Thermo Fisher). After 48 h, 10 µl of Cell Lysis buffer was added to a non-transfected cell triplicate and incubated for 45 min at 37 °C, 5% $CO_2$ to obtain maximum LDH activity. Afterward, 50 µl of each cell sample as well as the non-transfected cells for spontaneous LDH activity and maximum LDH activity, was transferred into a new 96-well plate and mixed with a 50 µl reaction mixture. Following 30 min of incubation at room temperature, 50 µl stop solution was added and the absorbance was measured at 490 and 680 nm using a Tecan M1000 plate reader. To determine LDH activity, the absorbance values for 680 nm were subtracted from that of 490 nm. The percentage of cytotoxicity based on maximum LDH activity was determined as follows: 100 × (cell sample LDH activity-spontaneous LDH activity)/(maximum LDH activity-spontaneous LDH activity).

## Fluid shear stress applications

We used the ibidi pump system (10905, Ibidi GmBH). Fluid shear stress levels were calibrated, and the imaging solution viscosity of the perfusion solution was determined according to the manufacturer's instructions. Depending on the level of fluid shear stress applied, perfusion set yellow-green (10964, for 5–30 dyn/cm²) or perfusion set white (10963, for 1–5 dyn/cm²) were used.

## Confocal and total internal reflection fluorescence microscopy (TIRFM) imaging

Images were acquired either on a Zeiss 780 confocal laser-scanning microscope equipped with an argon laser, a 561 nm diode pumped solid-state laser and a 633 nm HeNe laser using a C-Apochromat 40x/1.1 DICIII water immersion objective or on a Leica SP8 confocal laser-scanning microscope equipped with a white light laser (WLL, Leica) using either an HC PL APO CS2 40x/1.3 oil immersion objective or an HC PL APO CS2 63x/1.2 water immersion objective or on a STELLARIS 8 confocal laser-scanning microscope equipped with a white light laser (WLL, Leica) using either an HC PL APO CS2 40x/1.3 oil immersion objective or an HC PL APO CS2 63x/1.2 water immersion objective. Images of single cells were acquired at 8- or 16-bit depth, excited with 488 nm for a reporter, and 561 nm for tdTomato and jRCaMP1a excitation, respectively. To ensure fast image acquisition, we imaged small regions of interest within the field of view, recording a single z-plane over several minutes (pinhole size ranged from three Airy units to maximum pinhole size) using the maximum acquisition speed. When imaging was performed on the Leica SP8 confocal, bidirectional scanning with off-set correction was used to increase acquisition speed even more. Live imaging of cells was carried out either in Live Cell Imaging Solution (A14291DJ, Thermo Fisher) or culture medium without phenol red.

TIRFM images were obtained on an inverted widefield microscope (Ti Eclipse, Ti-TIRF-E Motorized TIRFM Illuminator, Nikon) operating in TIRFM mode, equipped with a 488 nm diode laser for excitation coupled with a 525/50 nm emission filter under ambient temperature conditions of 37 °C, 5% $CO_2$ using a 60x CFI Plan Apo TIRF NA 1.49 oil immersion objective (Nikon). Time-lapse images were recorded at 10 frames per second using an sCMOS camera (Neo sCMOS camera, Andor) combined with the Nikon NIS-Elements (Nikon) software to facilitate imaging. To minimize drift in focus across time and multiple regions, the perfect focus system (Nikon) was used to maintain axial focus.

Fluorescence imaging of the beating heart was performed using a Leica DMi8 combined with a CSU-X1 (Yokogawa, Tokyo, Japan) spinning at 10,000 rpm, two simultaneous cameras (TuCam Flash4.0, Hamamatsu, Shizuoka, Japan), and a water immersion objective (Leica 40X, N.A. 1.1). Embryos were mounted in 0.7% low melting-point agarose (Sigma-Aldrich) in a glass-bottom petri dish imaged at 28.5 °C at 100 frames per second. To stop the heart without unmounting embryos, 2 ml of 100 mM BDM (Sigma-Aldrich) and 0.4% tricaine (Sigma-Aldrich) solution was added to the mounting dish. Once the hearts were seen to stop beating, 2 ml of normal embryo media was added to the dish, and the embryos were imaged within 10 min. Where necessary, realignment of the beating heart was performed post-imaging using BeatSync2.0[84]. Embryos were heat-shocked at least 5 h before imaging on the day of the experiment. An MVX10 stereo microscope (Olympus), equipped with a CoolLED excitation light source (CoolLEDltd), using a 2x MV PLAPO objective (Olympus) was used for time-lapse widefield imaging of *Tg(hsp70:GenEPi)* and *Tg(kdrl:NLS-mCherry)* hearts and characterization of GenEPi responses during the heart beating cycle. ZEISS Efficient Navigation (ZEN) blue software was used to facilitate imaging.

The concentration of calcium is 1.8 mM in Live Cell Imaging Solution, 1.8 mM in the DMEM culture medium, 1.5 mM in IMDM culture medium and 33 nM in the E3 zebrafish medium.

## Atomic force microscopy (AFM)-based force spectroscopy and simultaneous confocal microscopy

Prior to the experiment, 5 μm diameter silica beads (Kisker Biotech) were glued to the free end of tipless cantilevers (CSC-37, Micromash HQ) using UV glue (Dymax) and cured under UV light for 20 min. Cantilevers with beads were plasma treated for 5 min using a plasma cleaner (Harrick Plasma) to ensure a clean surface, and subsequently mounted on a standard glass cantilever holder (JPK Instruments) of the AFM. Cells cultured on glass-bottom Petri dishes were kept at 37 °C using a Petri dish heater (JPK Instruments). For the mechanical stimulation, an AFM (CellHesion 200, JPK Instruments) was mounted on an inverted confocal microscope (Observer Z1, LSM 700, Zeiss). Cantilevers were calibrated using the thermal noise method[85]. Mechanical stimulation protocols were programmed using the JPK CellHesion software. During the mechanical stimulus, the AFM lowered the bead on the cantilever onto the cell with a speed of 10 μm/s until reaching the preset force, kept the preset force constant for 250 ms, and then retracted with a speed of 100 μm/s. Preset forces were applied in intervals from 100 to 400 nN with 50 nN increments, with the time between intervals ranging from 10–25 s.

Confocal imaging was performed using an inverted laser-scanning microscope (LSM 700, Zeiss) equipped with a 25x/0.8 LCI PlanApo water immersion objective (Zeiss). Time-lapse images were acquired with 100–300 ms time resolution and acquisition was initiated >10 s before the onset of the mechanical stimulus. Time-lapse images of $Ca^{2+}$ responses were analyzed using the built-in ZEN blue software.

## Patch-clamp electrophysiology

HEK293T cells were transfected with the plasmids using lipofectamine 2000 (Invitrogen) following the manufacturer's instructions. Forty-eight hours after transfection, whole-cell and cell-attached patch-clamp recordings were made on the cells at room temperature with the Axopatch-200B (Axon Instruments, Inc.) equipped with the Digidata 1550B and the pCLAMP 10.6 software (Molecular Devices, Sunnyvale, CA, USA). The tip resistance of recording glass pipettes was between 3 and 5 MΩ. The currents were sampled at 20 kHz and filtered at 2 kHz. The mechanical force was applied through a recording pipette using a Patchmaster-controlled pressure-clamp HSPC-1 device (ALA Scientific Instruments).

For whole-cell recordings, the external solution consisted of 133 mM NaCl, 3 mM KCl, 2.5 mM $CaCl_2$, 1 mM $MgCl_2$, 10 mM HEPES, and 10 mM glucose (pH 7.3 with NaOH). The pipette solution was composed of 133 mM CsCl, 1 mM $CaCl_2$, 1 mM $MgCl_2$, 5 mM EGTA, 10 mM HEPES, 4 mM MgATP, and 0.4 mM $Na_2GTP$ (pH 7.3 with CsOH).

For cell-attached recordings, the extracellular solutions were composed of 140 mM KCl, 1 mM $MgCl_2$, 10 mM glucose, and 10 mM HEPES (pH 7.3 with KOH). The pipette solutions consisted of 130 mM NaCl, 5 mM KCl, 1 mM $CaCl_2$, 1 mM $MgCl_2$, 10 mM TEA-Cl, and 10 mM HEPES (pH 7.3 with NaOH).

## In situ $Ca^{2+}$ titration of GenEPi

GenEPi-transfected HeLa cells were permeabilized using 150 μM β-escin (in 20 mM Na$^+$-HEPES, 140 mM KCl, 10 mM NaCl, 1 mM $MgCl_2$, pH 7.2) for 4 min. The solution was replaced with "zero free $Ca^{2+}$" solution (20 mM Na$^+$-HEPES, 140 mM KCl, 10 mM NaCl, 1 mM $MgCl_2$, 10 mM EGTA, pH 7.2) and various $Ca^{2+}$ concentrations (0.001, 0.01, 0.1, 1, 10, 50, 500, 10,000 μM free $Ca^{2+}$) were applied in the presence of 10 μM ionomycin and 4 μM thapsigargin. Free $Ca^{2+}$ concentrations were calculated using the two-chelators Maxchelator program[86].

Cells were examined with a Zeiss LSM 800 confocal microscope equipped with a 63x/1.4 Plan-Apochromat oil immersion objective and a 488 nm diode laser as an excitation light source. Emitted light was collected through Variable Secondary Dichroics (VSDs) onto a GaAsP-PMT detector. The fluorescence signal was monitored over an elliptical region of interest (ROI) in the plasma membrane using the ImageJ

program. Data obtained from 12 to 42 cells (from at least three independent experiments) was plotted and analyzed on GraphPad Prism 6. The fluorescence dynamic range ($(F_{max}-F_0)/F_0$ or $\Delta F/F_0$) was expressed as mean ± SEM. The $Ca^{2+}$ dissociation constant ($K_d$) and cooperativity (n) were obtained by fitting the data to the Hill equation.

## Live staining and immunochemistry

Live cell staining of cells was achieved using the Image-IT LIVE plasma membrane and nuclear labeling kit cell staining kit (I34406, Thermo Fisher), and the ER tracker red (E34250, Thermo Fisher) according to product specifications. For antibody staining, cells were fixed in 4% paraformaldehyde (15714-S, Lucerna Chem AG) for 5 min, washed with PBS, and blocked using Max Block blocking medium (15252, Active Motif) supplemented with 0.1% Triton X-100 (T8787, Sigma-Aldrich). Cells were then incubated with anti-GFP antibody (ab6673, Abcam, 1:100) or anti-Piezo1 antibody (ab82336, Abcam, 1:100) diluted in Max Block blocking medium. After several washing steps with PBS, the cells were incubated with goat anti-rabbit Alexa Fluor-633 (A-21071, 1:1000) or donkey anti-goat Alexa Fluor-633 (A-21082, 1:1000) as well as DAPI (62248, Thermo Fisher). Mouse embryos and mESCs were fixed in 10% paraformaldehyde (15714-S, Lucerna Chem AG) for 10 min, permeabilized with PBS supplemented with 0.1% Triton X-100 (T8787, Sigma-Aldrich) and blocked with 10% Donkey Serum (17-000-121, Jackson ImmunoResearch) in PBS supplemented with 0.1% Triton X-100. Samples were then incubated with anti-GFP antibody (ab6673, Abcam, 1:100), anti-mouse E-cadherin (AF748-SP, Techne AG. 1:100), anti-cardiac Troponin T antibody (ab8295, Abcam, 1:100), anti-smooth muscle myosin heavy chain II antibody (ab53219, Abcam, 1:100) or anti-CD31/PECAM-1 antibody (AF3628, R&D systems, 1:100) diluted in 10% Donkey Serum (17-000-121, Jackson ImmunoResearch) in PBS supplemented with 0.1% Triton X-100. After several washing steps with PBS, the samples were incubated with donkey anti-goat Alexa Fluor-594 (A-11058, 1:1000), goat anti-mouse Alexa Fluor-568 (A-11004, 1:1000), goat anti-rabbit Alexa Fluor-633 (A-21071, 1:1000), or donkey anti-goat Alexa Fluor-633 (A-21082, 1:1000) as well as DAPI (62248, Thermo Fisher).

## Zebrafish transgenesis and in vivo experiments

Experiments involving zebrafish were conducted in accordance with UK Home Office requirements (Animals Scientific Procedures Act 1986, project license P219D3ABD). All experiments were conducted up to 5 days post-fertilization.

To generate a GenEPi zebrafish transgenic line, Tol2-mediated transgenesis was used. About 30 pg of the hsp70:GenEPi plasmid and 120 pg of Tol2 transposase mRNA were injected into the blastomere of single WT/TL zebrafish embryos at the one-cell stage. The injected zebrafish embryos were heat-shocked at 16 hpf (hours post-fertilization) and those showing fluorescence were raised to adulthood. F1 founders were subsequently screened, and stable transgenic strains were established by selecting fish at F3 generation. Before imaging, *Tg(hsp70:GenEPi)* embryos were heat-shocked at 37.5 °C for 1 h to induce the expression of the GenEPi, and optically screened ~4–5 h post-heat-shock. All the zebrafish images were acquired at least 5 h post-heat-shock when a steady state of fluorescence is reached.

For the in vivo pharmacological intervention experiments, dechorionated zebrafish embryos were anesthetized in 0.2 mg/ml tricaine. Then, single embryos were sequentially transferred in aqueous solutions of (i) 10 μM Yoda1, (ii) 10 μM Yoda1 and 10 μM Ruthenium Red, (iii) 10 μM Ruthenium Red, and washed out in the original egg medium (E3 medium). After 5 min in each condition, each embryo was imaged in an MVX10 stereo microscope (Olympus), equipped with an X-Cite® 120Q excitation light source (Lumen Dynamics), using a 2x MV PLAPO objective (Olympus). ZEISS Efficient Navigation (ZEN) blue software was used to facilitate imaging.

For the longitudinal analysis of GenEPi expression post-heat-shock, individual embryos were placed in a glass-bottom 96-well plate (Greiner)

and imaged every 30 min using the Imaging Machine (Acquifer) with a 4X objective at 28 °C.

For membrane colocalization analysis, zebrafish embryos were either injected with mRFP RNA or live stained with 4 µM BODIPY TR Ceramide (Thermo Fisher, D7540) for 1 h or overnight before imaging.

Zebrafish heart beating measurements were manually measured over a MVX10 stereo microscope (Olympus) at specific timepoints, and data were further analyzed and plotted using GraphPad Prism 9.

For zebrafish immunostainings, embryos were dechorionated and fixed for 3 to 4 h at room temperature (RT) at the desired developmental stage in 4 g of PFA diluted in 100 ml of Fish Fix Buffer (1 L:1x PBS, 120 ml 1 M CaCl$_2$, 40 g sucrose). After fixation, embryos were washed in 1x PBS containing 0.1% Tween-20 (PBST). Embryos were then permeabilized in 1x PBST containing 0.5% Triton X-100 overnight at 4 °C. Embryos were blocked overnight at 4 °C in 1x PBST supplemented with 1% BSA, 0.5% Triton X-100, and 10% NGS. Antibody was used as follows: rabbit anti-GFP (A21312, Thermo Fisher) 1:400.

### Image processing and analysis

For the shear stress experiments, and time-lapse upon experiments with application of a chemical, the cells were automatically segmented using a MATLAB script. Briefly, the signals from the cytosolic tdTomato or jRCaMP1a were automatically identified, and high-intensity pixels were used to generate a mask. This mask was then applied to the time series images of the reporter and the cytosolic signal and single cell intensities were extracted for each time point. This information allowed us to get the traces for the intensiometric reporter response of each cell. Ca$^{2+}$ responses were expressed as fluorescence levels normalized to baseline ($F/F_O$). To obtain ($F/F_O$), we divided the fluorescence levels ($F$) by the baseline fluorescence of the cell (fluorescence of the first five frames, $F_O$).

During the AFM experiments, mechanical stimulation of cells with the cantilever caused cytosolic or membrane-bound fluorophores to move in or out of the confocal imaging plane, creating fluorescence artifacts. These artifacts were clearly distinguishable from Piezo1 receptor-mediated Ca$^{2+}$ influx, since they showed strong symmetry with stepwise increase and decrease of fluorescence. Any fluorescent signal that was greater than the artifact was classified as "response", anything below was defined as noise. The resulting signal trace was processed as described above. The duration of the signal was calculated by subtracting the first time point fluorescence signal is higher than the artifact from the time point signal goes back to the baseline. The baseline was calculated as the average fluorescence of 5 s preceding the stimulus.

Image analysis of the local and systemic responses of GenEPi in the differentiated cardiomyocytes experiments was performed using FiJi/ImageJ[87] (release 1.53f51). $F/F_O$ images were calculated using the built-in image calculator function ($F_O$ was the average of the first 5 frames of the time-lapse). The resulting $F/F_O$ images were pseudo-colored with the mpi-Inferno LUT to better demonstrate the fold increase differences. Quantitative data of the $F/F_O$ changes over time were collected from regions of interest or whole images using the plot z-axis profile (where z dimension was time, usually in milliseconds or seconds). Data were further analyzed and plotted using GraphPad Prism 9.

Image analysis of TIRFM imaging was performed with FiJi/ImageJ[87] (release 1.53f51). To segment GenEPi (Piezo1) clusters 8- or 16- bit single channel TIRFM images were masked by (i) subtracting the background (rolling ball radius of two pixels), (ii) applying a median filter (radius of two pixels), and (iii) thresholding (Otsu). The mask image was then subsequently watersheded, dilated, and watersheded again before the built-in analyze particles function of FiJi/ImageJ was used to obtain quantitative data, such as mean gray value, area, and perimeter in the original image. Single-pixel measurements due to masking artifacts were excluded from the subsequent analysis. Average camera noise

was subtracted from images that are displayed. When further filtering was used for illustration purposes, it is stated in the figure legends. To track individual GenEPi (Piezo1) membrane clusters Trackmate[88] was used with LoG detector (Laplacian of Gaussian filter), estimated object diameter of 1 µm, quality threshold of 1.0, and sub-pixel localization. The tracker used was the simple LAP tracker with a linking max distance of 1 µm, gap closing distance and frames of 1 µm and 20 frames, respectively. Data were saved as xml files and subsequently analyzed with MATLAB (R2021b) and @msdanalyser[89]. Data were further analyzed and plotted using GraphPad Prism 9.

To quantify fluorescence intensity changes in the zebrafish chemical modulation experiments, images were opened on FiJi/ImageJ[87] (release 1.53f51) and a rectangular region around the eye of each fish was manually selected. Mean intensity values were exported and analyzed/plotted using GraphPad Prism 9.

To quantify fluorescence intensity changes in the zebrafish longitudinal analysis experiments, images were opened on FiJi/ImageJ[87] (release 1.53f51) and a circular region of interest covering the whole embryo was manually selected. Mean intensity values for individual embryos were exported and analyzed/plotted using PlotTwist[90] (Shiny apps).

To quantify fluorescence intensity changes in the zebrafish heart beating experiments, time-lapse data were opened on FiJi/ImageJ[87] (release 1.53f51) and a rectangular region of interest covering the whole atrioventricular canal region of the zebrafish heart during the beating cycle was manually selected. Mean intensity values for individual embryos were exported along with background values from the yolk region and analyzed/plotted using GraphPad Prism.

Colocalization analyses were performed on FiJi/ImageJ[87] (release 1.53f51). Images were opened and a rectangular region covering one or more individual cells was manually selected. Then, the colocalization analysis Coloc2 plugin was used to implement and perform the pixel intensity correlation over space Pearson method. Individual Pearson's correlation coefficients were exported and analyzed/plotted using GraphPad Prism 9.

### Statistics and reproducibility

All data are expressed by means or median ± SEM. Sample sizes ($n$) are provided in the text or figure legend of each experiment. Each experiment has been repeated independently at least three times. Each data set was subjected to the Shapiro−Wilk normality test to determine whether the data set has a Gaussian distribution; $p > 0.05$ indicated it has a Gaussian distribution, and $p < 0.05$ indicated it did not. When all the compared datasets had Gaussian distribution, a two-tailed Student's $t$-test was applied to compare two independent datasets; with an $F$-test to compare variances. When $F$-test resulted in $p < 0.05$, Welch's correction was applied to the $t$-test. When more than two datasets were present with Gaussian distribution, one-way ANOVA was used to compare datasets, followed by Holm−Sidak's post hoc multiple comparisons test. When at least one of the compared datasets did not have a Gaussian distribution, Mann−Whitney test was applied to compare two independent datasets; and the Wilcoxon rank-sum test was applied when the datasets were paired. When more than two datasets were present without Gaussian distribution, Kruskal−Wallis test was applied, followed by Dunn's post hoc multiple comparisons test. For all statistics, either $p$ value number was reported or with n.s. = $p > 0.05$, * = $p < 0.05$, ** = $p < 0.01$, *** = $p < 0.001$, and **** = $p < 0.0001$. Mean-squared displacement analysis and linear fitting of the GenEPi (Piezo1) membrane clusters tracking dynamics were performed using MATLAB (R2021b) and @msdanalyser[89].

### Data availability

All the data that support the findings of this study are available from the corresponding author upon request. Raw data are provided in a Source Data file. Source data are provided with this paper.

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

## Acknowledgements

We thank members of the Pantazis group and F. Chatzidimitriou for their discussion and feedback. We thank W.P. Dempsey and M.A. Mohr for their comments on the manuscript. We would like to thank A.Y. Sonay for suggesting the name GenEPi for the Piezo1 sensor. This name was inspired by the traditional herbal liqueur Génépi, which originates from the alpine regions of France. This work was supported by the Swiss National Science Foundation (SNF grant. 31003A_144048 to P.P.), the European Union Seventh Framework Program (Marie Curie Career Integration Grant (CIG) no. 334552 to P.P.), the Biotechnology and Biological Sciences Research Council (BBSRC grant BB/T017929/1 to P.P.), the Royal Society Wolfson Research Merit Award to P.P., the Rubicon grant

from the Netherlands Organization for Scientific Research and a fellowship from the Peter und Traudl Engelhorn Stiftung to M.W., the European Molecular Biology Organization (EMBO; ALTF 424-2016 to B.M.G.), the NCCR Molecular Systems Engineering, the Wellcome Trust Project Grant 094385/Z/10/Z and BBSRC Project Grant (BB/MO2556X/1 to K.T.), the NIH Grant (RO1GM127876 to S.A.H.), and British Heart Foundation Intermediate Basic Science Research Fellowship (FS/17/56/32925 to N.H. and FS/17/2/32559 to J.S.). The Institute of Translational Medicine, Cellular and Molecular Physiology, University of Liverpool, is thanked for the use of the Zeiss LSM 800 confocal microscope. The Imperial College London and Leica Microsystems Imaging Hub is thanked for the use of the STELLARIS 8 confocal microscope. Figures 1a, 5a, b and Supplementary Figs. 15a, 21d were created with Biorender.com.

## Author contributions

S.Y. conceived and S.Y. and P.P. refined the idea. K.K and S.Y. designed, carried out, and analyzed experiments except for the following: K.T. designed the four GCaMPs (G1-G4); N.H. carried out the in situ affinity measurements; B.M.G. and D.J.M. designed and carried out the AFM-based force spectroscopy and simultaneous confocal microscopy; J.S. carried out patch-clamp electrophysiology; M.W. generated the dox-inducible mESC line; M.W and T.L. carried out differentiation experiments assisted by K.K. with the help of V.S.; S.S.T. carried out cell transfections; K.K designed and carried out the TIRFM imaging experiments with the help of D.L. and A.D.R.H.; D.J. generated the zebrafish transgenic line assisted by KK with the help of S.A.H.; C.V.P. and K.K. carried out the in vivo characterization experiments with the help of J.V.; K.K., S.Y., and P.P. wrote the manuscript and all authors contributed to editing the manuscript. P.P. supervised the project.

## Competing interests

The authors declare no competing interests.
