## [Peer Review File · Nature Communications]

REVIEWER COMMENTS

Reviewer #1 (Remarks to the Author):

To Authors,

Piezo1 acts as a mechanosensor in various cells. Activation of piezo1 induced by mechanical stimulation of cell membranes has been assessed by indirect methods, such as electrophysiological analysis or intracellular calcium imaging. If the activity of piezo1 induced by mechanical stimuli to the plasma membrane can be directly observed in living cells and organisms, it should be possible to more clearly assess the scene of piezo1 activation and its physiological role. This study developed a tool to directly monitor piezo1 activity. Authors have created GenEPI, a genetically-encoded fluorescent reporter for non-invasive optical monitoring of Piezo1-dependent activity. They demonstrate that GenEPI directly resolves Piezo1-dependent stimuli from the single-cell level to that of the entire organism. The fact that GenPi signaling reflects Piezo localization and activity has been verified by a wide variety of control experiments and is reliable. In particular, the GenPi signal is commendable in that it is clearly distinguishable from the intracellular Ca²⁺ transient caused by the 2nd messenger. This study will allow noninvasive direct monitoring of Piezo1 activity and will be a cooperative tool to study the mechanism of homeostasis via Piezo1-mediated mechanical feedback mechanisms. However, the interpretation of the vivo experimental system presented by the authors (Figures 4 and 5) is questionable.

Q1

In Fig.4G, authors show the fluorescence intensity profiles from ROI with GenPi activation in response to cardiomyocyte contraction after the addition of norepinephrine. The frequency of GenPi signaling is elevated (Fig. 4D), reflecting an increase in the beating rate of cardiomyocytes upon norepinephrine administration. I suspect that norepinephrine increases contractility (reduces end-systolic diameter) but does not alter the dilatability of microtissues. Why, then, does norepinephrine administration increase the amplitude of the GenPi signal? As shown in Supplementary Fig. 13, if the GenPi signal senses membrane extension (stretch or pulling), then norepinephrine administration may increase the frequency of the signal but not the amplitude.

Q2

In Supplementary Fig. 14, the authors show that myocyte contraction triggers the GenPi signal, which is thought to be elevated during the relaxation phase. If it can be shown that GenPi signal is elevated during relaxation (i.e., that the peak of contraction and the peak of GenPi signal are misaligned), the utility of GenPi will be more convincing.

Q3

The authors have generated a genetically modified zebrafish and have attempted to visualize piezo1 activity using GenPi in vivo. In Figure 2C, why is there no GenPi signal seen in the heart? In Figure 4, GenPi signal is observed reflecting contraction of cardiac tissue, but does this mean that Piezo1 expression in the heart is very low compared to all regions of the body?

Reviewer #2 (Remarks to the Author):

The authors cleverly created a fluorescent indicator of Piezo1 activation (GenEPI) by genetically fusing a calcium sensor near the intracellular side of the calcium-permeable pore. An impressive (heroic!) amount of data shows that GenEPI functionally behaves similar to wild-type Piezo1 channels, an important pre-requisite for GenEPI's intended applications. Although GenEPI undeniably appears as an innovative and interesting tool, its modest optical performance would limit its potential applications in its current design.

Major points:

GenEPI does produce detectable signals in response to Piezo1 stimuli in cells, but these signals remain small (1.6- to 2-fold fluorescence change) compared to traditional optical sensors such as GCaMPs (up to 10-fold fluorescence change), presumably because of the intrinsically low performance of parental

GCaMP-G4 (Fig 1C). Unfortunately, unless a better calcium indicator can be used in lieu of GCaMP-G4, this means that there is little room for improving GenEPI's dynamic range in the future.

Flow-dependent signals and ionomycin-induced signals from GenEPI are only marginally different, Fig 1C, 1.61- vs. 1.36-fold change (no statistical test provided), thus it seems difficult to believe that these flow-induced signals are "considerably higher" (L98) than ionomycin-induced signals or that GenEPI is a "highly specific" (title, L272) sensor of Piezo1 activity.

An important missing control would be to insert a pore mutation to abolish ion (calcium) permeation through GenEPI. A lack of signal from a non-conducting variant would definitely prove that signals observed under flow or in beating cardiomyocytes are specific to Piezo1, as ionomycin controls are done in absence of mechanical stimulation. In addition, negative controls using EGTA to chelate external calcium ions could also be useful to rule out contribution of intracellular store depletion.

The small signal-to-noise performance means that GenEPI's applications beyond in vitro systems are likely limited. Fig 5G suggests that this might be the case, as only faint activation in presence of Yoda1 vs. control is reported by GenEPI, while two-way ANOVA showed no significant differences of GenEPI fluorescence in presence / absence of Yoda1 / RR (Fig 5G). This data makes it difficult to believe the assertive claim that GenEPI enables "robust and reliable monitoring of Piezo1 activity in vivo" (L37).

Data from Fig 3 shows only one isolated spike of GenEPI fluorescence, but it is unclear if this sole signal is indeed due to sporadic activation of GenEPI or to transient displacement of fluorescent puncta into focus due to intrinsic mobility of GenEPI (Fig 3GH) and narrow depth of field of TIRF imaging.

The fact that GenEPI signals are independent on stimulus intensity (beyond a threshold) is concerning, as future experimentalists would likely want to test whether manipulating some variables correlates or not with increased / decreased Piezo1 activity.

Other points:

The first and last authors seem to have filed a patent for GenEPI. This would constitute a competing interest as per Nature' journals policy.

Except for electrophysiology and calcium dose-response experiments, calcium concentration of extracellular solution(s) (or composition of extracellular solutions) is not given.

There is no statistical analysis of data reported in Fig 1C

L48-49: the sentence should read something like "...and functional homologs [have been identified] in plants and invertebrates."

L63-64: the pore cannot reside within the cytosol, consider rephrasing

L151, L923: equal conductance does not mean equal ionic selectivity

L906: legend should include cells transfected with GenEPI

Figure 2P and Supplementary Fig. 8: information is missing about how data was obtained (data seems to be obtained from currents in whole-cell voltage clamped cells perfused with solution of varying Yoda1 concentrations and without mechanical stimulation).

Supplementary Fig. 9: at what pressure was the inactivation Tau compared?

Reviewer #3 (Remarks to the Author):

The recombinant protein GenEPI described in this work is composed of the mechanosensitive channel Piezo1 linked to calcium biosensor GCaMP6s RS-1 EF-4 variant (GCaMP-G4) with a flexible linker. In

this construct, GCaMP-G4 expression is restricted to the plasma membrane by its attachment to Piezo1, thus becoming sensitive to high calcium microdomains in the vicinity of the channel under the membrane, when the mechanosensor is activated.

The authors showed that mechanical stimuli which activate Piezo1 result in a fluorescence increase of the GCaMP-G4 moiety of GenEPi whereas addition of ionomycin, which causes a global Ca rise in the cytoplasm, did not. They also showed that GCaMP-G4 (expressed free in the cytoplasm) coexpressed with Piezo-1, responded both to mechanical stimuli and to ionomycin. It is intriguing why the same Ca biosensor does not respond to a global cytosolic Ca rise when it is linked to Piezo1.

The authors show that GenEPi reports the mechanical activation of Piezo1. However, they do not show whether local Ca microdomains caused by other Ca-permeable channels affect the GenEPi response. Thus, the specificity claimed in the title is not sufficiently demonstrated.

Piezo1 acts as a mechanosensor in many excitable cells (like those in Fig 4). In these cells, mechanical stimuli are associated with opening of voltage-activated calcium channels (like L-type channels, LTCCs), which cause local calcium microdomains under the plasma membrane. The spontaneously beating cardiomyocytes shown in Fig. 4 display membrane depolarizations, opening of LTCCs and Ca influx, Ca-induced Ca release through ryanodine receptors, followed by cell shortening in each beat. It is therefore important to check whether local Ca microdomains due to opening of LTCCs would contribute to the fluorescent readout of GenEPi, independent of the mechanical stimulus of Piezo1. This critical control seems to be lacking.

The authors stopped cardiomyocyte beating by treatment with the myosin inhibitor blebbistatin; this abolished the fluorescence change of GenEPi in the representative cell of Fig 4C. However, the response of GenEPi was not completely abrogated in all cardiomyocytes since the frequency of beating merely decreased (Fig S16). An experiment to test this point would be to block pharmacologically (i.e. with nifedipine) LTCCs to decrease a large % of LTCC current. This treatment should not change the optical readout of GenEPi if it is due only to Piezo1 opening.

Another possibility is to use blebbistatin or its analogs at concentrations that completely abrogate cardiomyocyte shortening. This myosin inhibitor does not preclude opening of LTCCs nor the occurrence of Ca transients. Test whether stopped cells, in which LTCCs continue to be activated, show any change in GenEPi fluorescence.

Minor questions:

- Indicate the concentration of blebbistatin used in Fig S15 in the figure legend.

- Expression of GenEPi in vivo after heat shock of zebrafish: fluorescence appeared 1 hour after heat shock. How much GenEPi is on the plasma membrane? Protein synthesis in the rough ER, processing in the Golgi apparatus, sorting in the secretory pathway and integration in the plasma membrane are processes which take time. Fig S16a seems to show intracellular staining compatible with the endoplasmic reticulum, in addition to plasma membrane.

- Fig S16: indicate how many hours after heat shock were the images acquired.

- The authors generated a zebrafish line expressing GenEPi to show that the biosensor works in vivo. Fig 5 shows the response to a chemical activator of Piezo1, but it would add relevance to the results to show the response to a mechanical stimulus. Since GenEPi is expressed in myotomes (Fig S16a), the authors could record fluorescence changes during spontaneous twitching of zebrafish. Two periods of calcium activity in the trunk and tail of 17–25 hpf developing zebrafish have been described (DOI: 10.1387/ijdb.103160cc). They are associated with twitching of skeletal muscle, resulting in coordinated movements, which could be followed by GenEPi.

REVIEWER COMMENTS

Reviewer #1 (Remarks to the Author):

Piezo1 acts as a mechanosensor in various cells. Activation of piezo1 induced by mechanical stimulation of cell membranes has been assessed by indirect methods, such as electrophysiological analysis or intracellular calcium imaging. If the activity of piezo1 induced by mechanical stimuli to the plasma membrane can be directly observed in living cells and organisms, it should be possible to more clearly assess the scene of piezo1 activation and its physiological role. This study developed a tool to directly monitor piezo1 activity. Authors have created GenEPi, a genetically-encoded fluorescent reporter for non-invasive optical monitoring of Piezo1-dependent activity. They demonstrate that GenEPi directly resolves Piezo1-dependent stimuli from the single-cell level to that of the entire organism. The fact that GenEPi signaling reflects Piezo localization and activity has been verified by a wide variety of control experiments and is reliable. In particular, the GenEPi signal is commendable in that it is clearly distinguishable from the intracellular Ca²⁺ transient caused by the 2nd messenger. This study will allow non-invasive direct monitoring of Piezo1 activity and will be a cooperative tool to study the mechanism of homeostasis via Piezo1-mediated mechanical feedback mechanisms.

We thank the reviewer for recognising the novelty and reliability of our work and we share their enthusiasm that the use of GenEPi will allow for “studying the mechanism of homeostasis via Piezo1-mediated mechanical feedback mechanisms”.

However, the interpretation of the vivo experimental system presented by the authors (Figures 4 and 5) is questionable.

Q1

In Fig.4G, authors show the fluorescence intensity profiles from ROI with GenPi activation in response to cardiomyocyte contraction after the addition of norepinephrine. The frequency of GenEPi signaling is elevated (Fig. 4D), reflecting an increase in the beating rate of cardiomyocytes upon norepinephrine administration. I suspect that norepinephrine increases contractility (reduces end-systolic diameter) but does not alter the dilatability of microtissues. Why, then, does norepinephrine administration increase the amplitude of the GenPi signal? As shown in Supplementary Fig. 13, if the GenPi signal senses membrane extension (stretch or pulling), then norepinephrine administration may increase the frequency of the signal but not the amplitude.

We thank the reviewer for their comment. As we show in Supplementary Fig 16, the membrane deformation caused by the contraction of attached cardiomyocytes leads to activation of GenEPi, an inherent quality of Piezo1 channels which respond to membrane tension (Cox et al., 2016; Lewis & Grandl, 2015; Syeda et al., 2015).

Norepinephrine has been shown to increase both the contraction rate and twitch amplitude of individual cardiac cells in a concentration dependent manner (Kaumann, 1987; Sakai et al., 1992). Indeed, we observe that this double effect of norepinephrine on the cardiomyocytes causes an increase of both the frequency and amplitude of GenEPi responses (Fig. 4D-F).

Q2

In Supplementary Fig. 14, the authors show that myocyte contraction triggers the GenEPi signal, which is thought to be elevated during the relaxation phase. If it can be shown that GenEPi signal is elevated during relaxation (i.e., that the peak of contraction and the peak of GenEPi signal are misaligned), the utility of GenEPi will be more convincing.

We appreciate the reviewer's comment and careful observation. In our analysis of contraction-triggered mechanical stimulation in cardiac microtissues we show that GenEPi activation is observed in GenEPi-expressing cells attached to autonomously beating cardiomyocytes. Thus, GenEPi activation aligns with the systolic phase (Supplementary Fig. 17) of the contracting cardiomyocytes when the latter apply the maximum contraction force. This contraction force subsequently leads to maximum membrane deformation of the attached GenEPi-expressing cell and triggers GenEPi response.

Q3

The authors have generated a genetically modified zebrafish and have attempted to visualize piezo1 activity using GenEPi in vivo. In Figure 2C, why is there no GenEPi signal seen in the heart? In Figure 4, GenEPi signal is observed reflecting contraction of cardiac tissue, but does this mean that Piezo1 expression in the heart is very low compared to all regions of the body?

Thank you for the important note on our GenEPi zebrafish system. To investigate GenEPi's ability to report Piezo1-dependent activity *in vivo*, we generated a zebrafish transgenic line that allows for conditional expression of GenEPi using the zebrafish heat-shock promoter hsp70-l. We decided to use this promoter to secure robust and temporally controlled GenEPi expression throughout the embryo, as was previously done by others (Halloran et al., 2000). We showed that GenEPi expression was systemic post-heat-shock and that fluorescence increase was significant compared to non-heat-shocked embryos (Fig. 5D-E). To demonstrate the robustness of our GenEPi zebrafish system, we have now included a longitudinal analysis of GenEPi expression post-heat-shock (Fig. 5C and Supplementary Fig. 19) and increased the number of analysed fish in Fig. 5E. In addition, we have now included images of the zebrafish heart at 3 and 1dpf (Fig. 5I and Supplementary Fig. 20) showing that GenEPi is expressed in both the myocardium and endocardium cells of the developing zebrafish heart post-heat-shock. We now demonstrate that GenEPi shows a highly stereotypical and mechanical stimuli-dependent activation during zebrafish heart beating which is

abolished when the heart is stopped with 2,3-Butanedione 2-monoxime (BDM) treatment (Fig. 5J-L). Notably, the heart rate of the heat-shocked embryos does not differ to non-heat-shocked ones when induced GenEPI reaches peak expression (5 hours post heat-shock) (Supplementary Fig. 21). We have now revised the text of our manuscript to include the new information (lines 283-292).

Reviewer #2 (Remarks to the Author):

The authors cleverly created a fluorescent indicator of Piezo1 activation (GenEPI) by genetically fusing a calcium sensor near the intracellular side of the calcium-permeable pore. An impressive (heroic!) amount of data shows that GenEPI functionally behaves similar to wild-type Piezo1 channels, an important pre-requisite for GenEPI's intended applications. Although GenEPI undeniably appears as an innovative and interesting tool, its modest optical performance would limit its potential applications in its current design.

We thank the reviewer for recognising the innovative nature of our sensor, the cleverness of our design and our “heroic” efforts to fully characterise GenEPI, our fluorescence indicator of Piezo1 activation.

Major points:

GenEPI does produce detectable signals in response to Piezo1 stimuli in cells, but these signals remain small (1.6- to 2-fold fluorescence change) compared to traditional optical sensors such as GCaMPs (up to 10-fold fluorescence change), presumably because of the intrinsically low performance of parental GCaMP-G4 (Fig 1C). Unfortunately, unless a better calcium indicator can be used in lieu of GCaMP-G4, this means that there is little room for improving GenEPI's dynamic range in the future.

We thank the reviewer for their constructive comment. In our systematic screening for a mechanosensitive indicator specific to Piezo1 activity, we tested 5 different genetically-encoded calcium indicators (GECIs) (see Fig. 1B) which were selected to fulfil 2 key requirements: (i) low affinity to calcium (K_d ranging from 610-6122 nM) to avoid “reporting” non-specific cytosolic calcium increases and (ii) high dynamic range (F_{max}/F_{min} ranging from 13.8-31) to easily distinguish their ON/OFF states. Our screening results (see Fig. 1C) showed that the fusion Piezo1-1XGSGG-GCaMP-G4 (GenEPI) fulfilled our initial requirements for a Piezo1 specific indicator (see Fig. 1D). Notably, the fusions of Piezo1 with GECIs of higher calcium affinity (e.g. GCaMP-G1 and GCaMP-G3) compared to other GECIs used in our screen did not yield a specific response, confirming our reasoning to use low affinity GECIs. Thus, GCaMP6s which is the parent protein of GCaMP-G4 and has much higher affinity to calcium (K_d 110nM) would not have been suitable for specifically monitoring Piezo1 activity.

Although the *in vitro* fluorescence dynamic range of GCaMP-G4 (GenEPI) is lower than its parent protein, GenEPI is able to specifically and accurately detect Piezo1

activity and has similar fluorescence characteristics to other commonly used *in vivo* biosensors (see **Revision Table 1**).

In the future, site-directed mutagenesis and high-throughput directed evolution approaches, which have proven successful for other biosensors (e.g. evolution of GCaMP3 to GCaMP6 (Chen et al., 2013; Dana et al., 2019; Zhang et al., 2021)), will allow us to refine even more the affinity of GenEPI to calcium and improve GenEPI response *in vivo*.

Revision Table 1. Comparison of GenEPI fluorescence dynamic range with other commonly used *in vivo* biosensors.

Biosensor	Biological process	Fluorescence dynamic range $\Delta F/F_0$	Reference
GenEPI (Ca²⁺)	GenEPI activation in response to cardiomyocyte contraction after the addition of 100 nM norepinephrine	0.31	This paper (Fig. 4F)
GCaMP6s (Ca²⁺)	Ca ²⁺ release from single action potential	0.28	Chen et al., Nature, 2013 (Fig. 1B)
iGluSnFR (glutamate)	Glutamate release from single presynaptic terminals	0.80	Helassa et al., PNAS, 2018 (Fig. 2A)
iGABASnFR (GABA)	Extracellular GABA transients in transfected acute brain slices using 2 mM Ca ²⁺ stimulation	0.33	Marvin et al., Nature Methods, 2019 (Fig. 2C)

Flow-dependent signals and ionomycin-induced signals from GenEPI are only marginally different, Fig 1C, 1.61- vs. 1.36-fold change (no statistical test provided), thus it seems difficult to believe that these flow-induced signals are “considerably higher” (L98) than ionomycin-induced signals or that GenEPI is a “highly specific” (title, L272) sensor of Piezo1 activity.

We thank the reviewer for their comment. We would like to point out that the detailed statistical information for our systematic screen is presented in Supplementary Table 3 (page 53 of original submission file, lines 904 and 1269) and shows that the differences mentioned by the reviewer are statistically significant. To ease the interpretation of the screening results we have now included a summary in Fig. 1D. Briefly, the fusion variant Piezo1-1XGSGG-GCaMP-G4 (GenEPI) was the only variant that showed both significantly higher response to the Piezo1-dependent stimuli (shear stress) and significantly lower response to the non-Piezo1 specific stimuli (ionomycin) compared to cytosolic responses, respectively. Notably, the duplication of the flexible linker unit (GSGG) was sufficient to abolish the specificity to Piezo1-dependent stimuli (see Fig. 1C, GCaMP-G4) which confirms our hypothesis that the specificity of GenEPI

to report Piezo1-dependent signals is the result of our design using both a flexible linker of defined length and a low calcium affinity genetically-encoded indicator. Hence, we believe “that this design principle of GenEPi can serve as a blueprint for developing and engineering optical reporters of other ion channels without affecting their function” (lines 309-311).

An important missing control would be to insert a pore mutation to abolish ion (calcium) permeation through GenEPi. A lack of signal from a non-conducting variant would definitely prove that signals observed under flow or in beating cardiomyocytes are specific to Piezo1, as ionomycin controls are done in absence of mechanical stimulation. In addition, negative controls using EGTA to chelate external calcium ions could also be useful to rule out contribution of intracellular store depletion.

We thank the reviewer for suggesting excellent control experiments to strengthen our manuscript. To further confirm the specificity of GenEPi response, we chelated extracellular calcium with EGTA as previously described by Syeda et al. (Syeda et al., 2015). We reasoned that chelation of extracellular calcium will not elicit any GenEPi response as no calcium would flow through the channel. GenEPi fluorescence significantly increased in response to the Yoda1 treatment (Yoda1), whereas chelation of extracellular calcium did not increase GenEPi fluorescence both in absence (EGTA) and presence of Yoda1 (Yoda1+EGTA) (Supplementary Fig. 7). Altogether, we demonstrate that GenEPi shows a Piezo1 specific response which is affected by extracellular calcium chelation, confirming that its fluorescence increase is due to the calcium flow upon Piezo1 channel opening.

Loss-of-function missense mutations of Piezo1 have been linked to several diseases, such as generalized lymphatic dysplasia (Fotiou et al., 2015; Lukacs et al., 2015), and bicuspid aortic valve (Faucherre et al., 2020). Recently, the loss-of-function S217L (Serine 217 to Leucine) missense mutation of Piezo1 has been shown to exhibit reduced plasma membrane trafficking, reduced stability and higher ubiquitination compared to wt-Piezo1 (Zhou et al., 2021). To further test the functionality of GenEPi and confirm that it can recapitulate Piezo1 disease conditions, we used site-directed mutagenesis to generate a GenEPi-S217L variant and tested whether the GenEPi-S217L mutant exhibits similar subcellular localization. Non-mutated GenEPi localizes both in the plasma membrane and the ER reflecting the subcellular localization of wt-Piezo1 (Supplementary Note 1-Fig. 1d), whereas GenEPi-S217L co-localizes almost exclusively with the ER tracker (Supplementary Note 1-Fig. 1d-e) like Piezo1-S217L (Zhou et al., 2021). In contrast to non-mutated GenEPi, GenEPi-S217L does not respond to treatment with 10 μ M of Yoda1 (Supplementary Note 1-Fig. 1f). Interestingly, GenEPi-S217L shows a significant decrease in its fluorescence upon agonist stimulation. Overall, we demonstrate that GenEPi reflects the subcellular localization of wt-Piezo1 and can recapitulate Piezo1 disease conditions which are linked to loss-of-function mutations (lines 97-98, 1198-1213).

The small signal-to-noise performance means that GenEPI's applications beyond in vitro systems are likely limited. Fig 5G suggests that this might be the case, as only faint activation in presence of Yoda1 vs. control is reported by GenEPI, while two-way ANOVA showed no significant differences of GenEPI fluorescence in presence / absence of Yoda1 / RR (Fig 5G). This data makes it difficult to believe the assertive claim that GenEPI enables "robust and reliable monitoring of Piezo1 activity in vivo" (L37).

We thank the reviewer for pointing out that the statistical significance of the results in the original Fig. 5G were difficult to be interpreted. To demonstrate the statistical significance in a more effective way and to strengthen our claims, we have now changed the way the statistical significance is presented between compared groups and have increased the number of sequentially treated zebrafish (Fig. 5G-H). Briefly, we show that GenEPI fluorescence increases significantly when fish are transferred to E3 medium containing Yoda1 and decreases when fish are transferred sequentially to E3 medium containing ruthenium red. Overall, we demonstrate that GenEPI has similar responses to our *in vitro* analyses, and we prove its robust and reliable monitoring of Piezo1 activity across scales.

In addition, we have now included images of the zebrafish heart at 3 and 1dpf (Fig. 5I and Supplementary Fig. 20) showing that GenEPI is expressed in both the myocardium and endocardium cells of the developing zebrafish heart post-heat-shock. We now demonstrate that GenEPI shows a highly stereotypical and mechanical stimuli-dependent activation during zebrafish heart beating which is abolished when the heart is stopped with 2,3-Butanedione 2-monoxime (BDM) treatment (Fig. 5J-L). Notably, the heart rate of the heat-shocked embryos does not differ to non-heat-shocked ones when induced GenEPI fluorescence reaches peak expression (5 hours post heat-shock) (Supplementary Fig. 19). We have now revised the text of our manuscript to include the new information (line 283-292).

Data from Fig 3 shows only one isolated spike of GenEPI fluorescence, but it is unclear if this sole signal is indeed due to sporadic activation of GenEPI or to transient displacement of fluorescent puncta into focus due to intrinsic mobility of GenEPI (Fig 3GH) and narrow depth of field of TIRF imaging.

We thank the reviewer for their comment on our TIRFM data. In our time-lapse TIRFM experimental setup, we used a 488nm laser at an incident angle of 62°, which resulted in an evanescent field penetration depth of ~188nm that was sufficient to secure the capturing of all membrane GenEPI dynamics. To strengthen our analysis and show the utility of GenEPI to study the activity of Piezo1 on the cell membrane, we have now included in Supplementary Fig. 13 an analysis of 49 GenEPI clusters of the cell shown in Fig. 3I which demonstrate the highly dynamic activity of GenEPI (Piezo1) clusters

previously reported also by others (Ellefsen et al., 2019; Holt et al., 2021; Pathak et al., 2014).

The fact that GenEPi signals are independent on stimulus intensity (beyond a threshold) is concerning, as future experimentalists would likely want to test whether manipulating some variables correlates or not with increased/decreased Piezo1 activity.

We thank the reviewer for their comment. As it has been previously shown, Piezo1 contains mechanically sensitive domains (Wu et al., 2016) and its activation by mechanical forces is threshold-dependent and directly related to the direction of the force applied (Gaub & Muller, 2017). We demonstrate that this inherent characteristic of the channel is preserved within GenEPi which showed a threshold-dependent activation upon mechanical stimulation with an AFM cantilever (Fig. 2J). Due to the highly cooperative Ca^{2+} sensing mechanism of the GCaMP, GenEPi is able to robustly report Ca^{2+} influx upon channel opening (activation of Piezo1) with high spatiotemporal resolution. Notably, in our norepinephrine experiments we show that relative comparisons of GenEPi activity are possible. Indeed, we observed an increase of the amplitude of GenEPi responses when higher concentrations of norepinephrine were used (see response to reviewer #1). In addition, we now demonstrate that GenEPi shows a dose-dependent response to the Piezo1-specific agonist Yoda1 (Supplementary Fig. 6), enabling comparative studies upon different treatments.

Other points:

The first and last authors seem to have filed a patent for GenEPi. This would constitute a competing interest as per Nature' journals policy.

The patent application filed in 2016 (EP 2016/191538) has been discontinued.

Except for electrophysiology and calcium dose-response experiments, calcium concentration of extracellular solution(s) (or composition of extracellular solutions) is not given.

We have now changed the manuscript text and included the concentration of calcium in the different solutions used during image acquisition (lines 549-551).

There is no statistical analysis of data reported in Fig 1C.

In order to simplify the presentation of Fig. 1C, we chose to present the detailed statistical information for our systemic screen in Supplementary Table 3 (page 53 of original submission file, line 1269). We have now included a summary of the systematic screening results in Fig. 1D to ease the interpretation of the data.

L48-49: the sentence should read something like "...and functional homologs [have been identified] in plants and invertebrates."

L63-64: the pore cannot reside within the cytosol, consider rephrasing
L151, L923: equal conductance does not mean equal ionic selectivity
L906: legend should include cells transfected with GenEPI

We thank the reviewer for the careful reading of our manuscript. We have now changed the text according to the reviewer's suggestions.

Figure 2P and Supplementary Fig. 8: information is missing about how data was obtained (data seems to be obtained from currents in whole-cell voltage clamped cells perfused with solution of varying Yoda1 concentrations and without mechanical stimulation).

In both figures, the whole-cell currents were recorded with the bathing solution in which different concentrations of Yoda1 was added, in the absence of any form of mechanical stimulation. We have now included the information to the figure legend (lines 1072, 1395-1396).

Supplementary Fig. 9: at what pressure was the inactivation Tau compared?

In Supplementary Fig. 11 (old Supplementary Fig. 9), the pressure at which the inactivation Tau was compared is -60 mmHg. We have now added the description in the figure legend (line 1405).

Reviewer #3 (Remarks to the Author):

The recombinant protein GenEPI described in this work is composed of the mechanosensitive channel Piezo1 linked to calcium biosensor GCaMP6s RS-1 EF-4 variant (GCaMP-G4) with a flexible linker. In this construct, GCaMP-G4 expression is restricted to the plasma membrane by its attachment to Piezo1, thus becoming sensitive to high calcium microdomains in the vicinity of the channel under the membrane, when the mechanosensor is activated.

The authors showed that mechanical stimuli which activate Piezo1 result in a fluorescence increase of the GCaMP-G4 moiety of GenEPI whereas addition of ionomycin, which causes a global Ca rise in the cytoplasm, did not. They also showed that GCaMP-G4 (expressed free in the cytoplasm) co-expressed with Piezo-1, responded both to mechanical stimuli and to ionomycin. It is intriguing why the same Ca biosensor does not respond to a global cytosolic Ca rise when it is linked to Piezo1.

We thank the reviewer for their interest in our work. Our efforts to generate an optical reporter of Piezo1-dependent activity were based on the calcium microdomain hypothesis (Clapham, 2007). This hypothesis states that upon channel opening the concentration of calcium in the vicinity of the ion permeating pore would significantly increase in comparison with the closed channel (see **Revision Fig. 1**). Therefore, we

decided to target the ion permeating channel of Piezo1 with a genetically-encoded calcium indicator (GECI).

Revision Fig. 1. The microdomain hypothesis for Piezo1 which inspired our work on GenEPI.

In our systematic screening, we tested 5 different GECIs (see Fig. 1B) which were selected to fulfil 2 key requirements: (i) low affinity to calcium (K_d ranging from 610-6122 nM) to avoid “reporting” non-specific cytosolic calcium increases and (ii) high dynamic range (F_{max}/F_{min} ranging from 13.8-31) to easily distinguish their ON/OFF states. In addition, we tested 2 different types of flexible linkers (1X or 2X of GSGG) to secure the proper folding of the Piezo1 protein and to not alter the functionality of the channel. Our screening results (see Fig. 1C) showed that the fusion Piezo1-1XGSGG-GCaMP-G4 (GenEPI) fulfilled our initial requirements for a Piezo1 specific indicator and showed a robust response to the Piezo1-dependent stimuli (shear stress) (see revised Fig. 1D). Notably, fusions of Piezo1 with GECIs of higher calcium affinity (e.g. GCaMP-G1 and GCaMP-G3) compared to the others used in our screening did not yield a specific response, confirming our reasoning to use low affinity GECIs. Furthermore, the duplication of the flexible linker unit (GSGG) was sufficient to abolish the specificity to Piezo1-dependent stimuli (see Fig. 1C, GCaMP-G4) which confirms our microdomain hypothesis and demonstrates that the specificity of GenEPI to report Piezo1-dependent signals is the result of our design using both a flexible linker of defined length and a low calcium affinity GECI. Hence, we believe “that this design principle of GenEPI can serve as a blueprint for developing and engineering optical reporters of other ion channels without affecting their function” (lines 309-311).

The authors show that GenEPI reports the mechanical activation of Piezo1. However, they do not show whether local Ca microdomains caused by other Ca-permeable channels affect the GenEPI response. Thus, the specificity claimed in the title is not sufficiently demonstrated.

Piezo1 acts as a mechanosensor in many excitable cells (like those in Fig 4). In these cells, mechanical stimuli are associated with opening of voltage-activated calcium channels (like L-type channels, LTCCs), which cause local calcium microdomains under the plasma membrane. The spontaneously

beating cardiomyocytes shown in Fig. 4 display membrane depolarizations, opening of LTCCs and Ca influx, Ca-induced Ca release through ryanodine receptors, followed by cell shortening in each beat. It is therefore important to check whether local microdomains due to opening of LTCCs would contribute to the fluorescent readout of GenEPi, independent of the mechanical stimulus of Piezo1. This critical control seems to be lacking.

The authors stopped cardiomyocyte beating by treatment with the myosin inhibitor blebbistatin; this abolished the fluorescence change of GenEPi in the representative cell of Fig 4C. However, the response of GenEPi was not completely abrogated in all cardiomyocytes since the frequency of beating merely decreased (Fig S16). An experiment to test this point would be to block pharmacologically (i.e. with nifedipine) LTCCs to decrease a large % of LTCC current. This treatment should not change the optical readout of GenEPi if it is due only to Piezo1 opening.

We thank the reviewer for their constructive comment on our cardiac microtissues experiments and for suggesting an excellent control experiment to strengthen our analysis. To further demonstrate that other channels, such as L-type channels, do not contribute to the fluorescence increase of GenEPi, we have now treated differentiated cardiac microtissues with 100 nM nifedipine which is known to block most L-type channels and has a strong negative inotropic effect (Brixius et al., 2005; Mannhardt et al., 2017; Xi et al., 2010). As shown in the revised Fig. 4H-J, nifedipine treatment indeed abolished the beating of cardiomyocytes within the differentiated cardiac microtissues, suggesting that L-type channels were effectively blocked. Yet, the baseline fluorescence of the GenEPi-expressing cells (Fig. 4K, L) is not affected by blocking L-type channels. These data show that other calcium channels, such as L-types channels, do not affect the Piezo1-dependent functional readout of GenEPi (lines 232-239).

In addition, we show in Supplementary Note 2 that the plasma membrane localization of GCaMP-G4 alone does not confer functional specificity to mechanical stimulation. Briefly, we attached the membrane targeting sequence of the protein tyrosine kinase Lck to the calcium indicator GCaMP-G4 (Lck-GCaMP-G4) and subsequently exposed Lck-GCaMP-G4 and Piezo1 co-transfected cells to fluid shear stress and ionomycin. As shown in Supplementary Note 2-Fig.1 the response of Lck-GCaMP-G4 to fluid shear stress was significantly lower than that of GenEPi, while showing a pronounced response to ionomycin (Supplementary Note 2-Fig. 1). Hence, membrane localization alone is not sufficient to acquire functional specificity to mechanical stimuli and supports our microdomain hypothesis.

Minor questions:

Indicate the concentration of blebbistatin used in Fig S15 in the figure legend.

We thank the reviewer for their careful notes on our manuscript. We have now added the concentration of blebbistatin used in Supplementary Fig. 18 (old Supplementary Fig. 15).

Expression of GenEPi in vivo after heat shock of zebrafish: fluorescence appeared 1 hour after heat shock. How much GenEPi is on the plasma membrane? Protein synthesis in the rough ER, processing in the Golgi apparatus, sorting in the secretory pathway and integration in the plasma membrane are processes which take time. Fig S16a seems to show intracellular staining compatible with the endoplasmic reticulum, in addition to plasma membrane.

Fig S16: indicate how many hours after heat shock were the images acquired.

We thank the reviewer for their remark regarding our zebrafish GenEPi system. We have now adjusted the schematic of the heat-shock protocol in Fig. 5B to clearly state the timing of the events which are described in detail in the Online Methods (see lines 644-647). To further support our findings, we have now included a longitudinal analysis of heat-shocked *Tg(hsp70:GenEPi)* zebrafish which shows that the maximum levels of GenEPi expression are observed after 4-5 hours post heat-shock (line 255-257).

In addition, we would like to point out that all the images were acquired at least 5 hours post-heat-shock when a steady state of fluorescence was reached. We have now included this information in our revised manuscript (see lines 646-647). Finally, we changed the legend of Supplementary Fig. 20 (old Supplementary Fig. 18) and indicated at which stage the images were acquired (see for example line 1479).

The authors generated a zebrafish line expressing GenEPi to show that the biosensor works in vivo. Fig 5 shows the response to a chemical activator of Piezo1, but it would add relevance to the results to show the response to a mechanical stimulus. Since GenEPi is expressed in myotomes (Fig S16a), the authors could record fluorescence changes during spontaneous twitching of zebrafish. Two periods of calcium activity in the trunk and tail of 17–25hpf developing zebrafish have been described (DOI: 10.1387/ijdb.103160cc). They are associated with twitching of skeletal muscle, resulting in coordinated movements, which could be followed by GenEPi.

We thank the reviewer for suggesting another set of excellent *in vivo* experiments to strengthen our analysis. As we have now included images of heat-shocked *Tg(hsp70:GenEPi)* zebrafish at 3 and 1dpf (Fig. 5I and Supplementary Fig. 20) which show the expression of GenEPi in the developing zebrafish heart, we reasoned to focus our analysis on zebrafish heart beating. In addition to the pharmacological *in vivo* validation in our original manuscript (Fig. 5), we share now a mechanical, autonomous *in vivo* validation of GenEPi included in our revised manuscript. We demonstrate that GenEPi shows a highly stereotypical and mechanical stimuli-dependent activation during zebrafish heart beating which is abolished when the heart is stopped with 2,3-Butanedione 2-monoxime (BDM) treatment (Fig. 5J-L). Notably,

the heart rate of the heat-shocked embryos does not differ to non-heat-shocked ones when induced GenEPI reaches peak expression (5 hours post heat-shock) (Supplementary Fig. 19). We have now revised the text of our manuscript to include the new information (line 283-292).

References

- Brixius, K., Gross, T., Tossios, P., Geissler, H. J., Mehlhorn, U., Schwinger, R. H., & Hekmat, K. (2005). Increased vascular selectivity and prolonged pharmacological efficacy of the L-type Ca²⁺ channel antagonist lercanidipine in human cardiovascular tissue. *Clin Exp Pharmacol Physiol*, 32(9), 708-713. <https://doi.org/10.1111/j.1440-1681.2005.04265.x>
- Chen, T. W., Wardill, T. J., Sun, Y., Pulver, S. R., Renninger, S. L., Baohan, A., Schreiter, E. R., Kerr, R. A., Orger, M. B., Jayaraman, V., Looger, L. L., Svoboda, K., & Kim, D. S. (2013). Ultrasensitive fluorescent proteins for imaging neuronal activity. *Nature*, 499(7458), 295-300. <https://doi.org/10.1038/nature12354>
- Clapham, D. E. (2007). Calcium signaling. *Cell*, 131(6), 1047-1058. <https://doi.org/10.1016/j.cell.2007.11.028>
- Cox, C. D., Bae, C., Ziegler, L., Hartley, S., Nikolova-Krstevski, V., Rohde, P. R., Ng, C.-A., Sachs, F., Gottlieb, P. A., & Martinac, B. (2016). Removal of the mechanoprotective influence of the cytoskeleton reveals PIEZO1 is gated by bilayer tension. *Nat Commun*, 7(1), 10366. <https://doi.org/10.1038/ncomms10366>
- Dana, H., Sun, Y., Mohar, B., Hulse, B. K., Kerlin, A. M., Hasseman, J. P., Tsegaye, G., Tsang, A., Wong, A., & Patel, R. (2019). High-performance calcium sensors for imaging activity in neuronal populations and microcompartments. *Nat Methods*, 16(7), 649-657.
- Ellefsen, K. L., Holt, J. R., Chang, A. C., Nourse, J. L., Arulmoli, J., Mekhdjian, A. H., Abuwarda, H., Tombola, F., Flanagan, L. A., Dunn, A. R., Parker, I., & Pathak, M. M. (2019). Myosin-II mediated traction forces evoke localized Piezo1-dependent Ca²⁺ flickers. *Communications Biology*, 2(1). <https://doi.org/10.1038/s42003-019-0514-3>
- Faucherre, A., Moha Ou Maati, H., Nasr, N., Pinard, A., Theron, A., Odelin, G., Desvignes, J. P., Salgado, D., Collod-Bérout, G., Avierinos, J. F., Lebon, G., Zaffran, S., & Jopling, C. (2020). Piezo1 is required for outflow tract and aortic valve development. *J Mol Cell Cardiol*, 143, 51-62. <https://doi.org/10.1016/j.yjmcc.2020.03.013>
- Fotiou, E., Martin-Almedina, S., Simpson, M. A., Lin, S., Gordon, K., Brice, G., Atton, G., Jeffery, I., Rees, D. C., Mignot, C., Vogt, J., Homfray, T., Snyder, M. P., Rockson, S. G., Jeffery, S., Mortimer, P. S., Mansour, S., & Ostergaard, P. (2015). Novel mutations in PIEZO1 cause an autosomal recessive generalized lymphatic dysplasia with non-immune hydrops fetalis. *Nat Commun*, 6, 8085. <https://doi.org/10.1038/ncomms9085>
- Gaub, B. M., & Muller, D. J. (2017). Mechanical Stimulation of Piezo1 Receptors Depends on Extracellular Matrix Proteins and Directionality of Force. *Nano Lett*, 17(3), 2064-2072. <https://doi.org/10.1021/acs.nanolett.7b00177>
- Halloran, M. C., Sato-Maeda, M., Warren, J. T., Su, F., Lele, Z., Krone, P. H., Kuwada, J. Y., & Shoji, W. (2000). Laser-induced gene expression in specific cells of transgenic zebrafish. *Development*, 127(9), 1953-1960.
- Helassa, N., Dürst, C. D., Coates, C., Kerruth, S., Arif, U., Schulze, C., Wiegert, J. S., Geeves, M., Oertner, T. G., & Török, K. (2018). Ultrafast glutamate sensors resolve high-frequency release at Schaffer collateral synapses. *Proc Natl Acad Sci U S A*, 115(21), 5594-5599. <https://doi.org/10.1073/pnas.1720648115>
- Holt, J. R., Zeng, W.-Z., Evans, E. L., Woo, S.-H., Ma, S., Abuwarda, H., Loud, M., Patapoutian, A., & Pathak, M. M. (2021). Spatiotemporal dynamics of PIEZO1 localization controls keratinocyte migration during wound healing. *Elife*, 10, e65415. <https://doi.org/10.7554/eLife.65415>

- Kaumann, A. J. (1987). Adrenaline and noradrenaline increase contractile force of human ventricle through both beta 1- and beta 2-adrenoceptors. *Biomed Biochim Acta*, 46(8-9), S411-416.
- Lewis, A. H., & Grandl, J. (2015). Mechanical sensitivity of Piezo1 ion channels can be tuned by cellular membrane tension. *Elife*, 4. <https://doi.org/10.7554/eLife.12088>
- Lukacs, V., Mathur, J., Mao, R., Bayrak-Toydemir, P., Procter, M., Cahalan, S. M., Kim, H. J., Bandell, M., Longo, N., Day, R. W., Stevenson, D. A., Patapoutian, A., & Krock, B. L. (2015). Impaired PIEZO1 function in patients with a novel autosomal recessive congenital lymphatic dysplasia. *Nat Commun*, 6, 8329. <https://doi.org/10.1038/ncomms9329>
- Mannhardt, I., Eder, A., Dumotier, B., Prondzynski, M., Krämer, E., Traebert, M., Söhren, K. D., Flenner, F., Stathopoulou, K., Lemoine, M. D., Carrier, L., Christ, T., Eschenhagen, T., & Hansen, A. (2017). Blinded Contractility Analysis in hiPSC-Cardiomyocytes in Engineered Heart Tissue Format: Comparison With Human Atrial Trabeculae. *Toxicol Sci*, 158(1), 164-175. <https://doi.org/10.1093/toxsci/kfx081>
- Marvin, J. S., Shimoda, Y., Magloire, V., Leite, M., Kawashima, T., Jensen, T. P., Kolb, I., Knott, E. L., Novak, O., Podgorski, K., Leidenheimer, N. J., Rusakov, D. A., Ahrens, M. B., Kullmann, D. M., & Looger, L. L. (2019). A genetically encoded fluorescent sensor for in vivo imaging of GABA. *Nat Methods*, 16(8), 763-770. <https://doi.org/10.1038/s41592-019-0471-2>
- Pathak, M. M., Nourse, J. L., Tran, T., Hwe, J., Arulmoli, J., Le, D. T. T., Bernardis, E., Flanagan, L. A., & Tombola, F. (2014). Stretch-activated ion channel Piezo1 directs lineage choice in human neural stem cells. *Proceedings of the National Academy of Sciences*, 111(45), 16148-16153. <https://doi.org/10.1073/pnas.1409802111>
- Sakai, M., Danziger, R. S., Xiao, R. P., Spurgeon, H. A., & Lakatta, E. G. (1992). Contractile response of individual cardiac myocytes to norepinephrine declines with senescence. *Am J Physiol*, 262(1 Pt 2), H184-189. <https://doi.org/10.1152/ajpheart.1992.262.1.H184>
- Syeda, R., Xu, J., Dubin, A. E., Coste, B., Mathur, J., Huynh, T., Matzen, J., Lao, J., Tully, D. C., Engels, I. H., Petrassi, H. M., Schumacher, A. M., Montal, M., Bandell, M., & Patapoutian, A. (2015). Chemical activation of the mechanotransduction channel Piezo1. *Elife*, 4. <https://doi.org/10.7554/eLife.07369>
- Wu, J., Goyal, R., & Grandl, J. (2016). Localized force application reveals mechanically sensitive domains of Piezo1. *Nat Commun*, 7, 12939. <https://doi.org/10.1038/ncomms12939>
- Xi, J., Khalil, M., Shishechian, N., Hannes, T., Pfannkuche, K., Liang, H., Fatima, A., Haustein, M., Suhr, F., Bloch, W., Reppel, M., Sarić, T., Wernig, M., Jänisch, R., Brockmeier, K., Hescheler, J., & Pillekamp, F. (2010). Comparison of contractile behavior of native murine ventricular tissue and cardiomyocytes derived from embryonic or induced pluripotent stem cells. *Faseb j*, 24(8), 2739-2751. <https://doi.org/10.1096/fj.09-145177>
- Zhang, Y., Rózsa, M., Liang, Y., Bushey, D., Wei, Z., Zheng, J., Reep, D., Broussard, G. J., Tsang, A., & Tsegaye, G. (2021). Fast and sensitive GCaMP calcium indicators for imaging neural populations. *bioRxiv*, 2021.2011.2008.467793.
- Zhou, Z., Li, J. V., Martinac, B., & Cox, C. D. (2021). Loss-of-Function Piezo1 Mutations Display Altered Stability Driven by Ubiquitination and Proteasomal Degradation. *Front Pharmacol*, 12, 766416. <https://doi.org/10.3389/fphar.2021.766416>

REVIEWER COMMENTS

Reviewer #1 (Remarks to the Author):

With appropriate additional experimentation, the question has been answered. In the future, I hope to use such tools to clarify the contribution of piezo1 caused by different hemodynamic loads (e.g., left ventricular hypertrophy caused by systolic loading and cardiodilation caused by diastolic loading).

Reviewer #2 (Remarks to the Author):

[Privately signs off in the 'Remarks to the Editor']

Reviewer #3 (Remarks to the Author):

The authors have added further results to address the questions raised in the review and have elucidated some points. However, some of the data and arguments proposed do not address and clarify the key criticisms:

From the first review "The authors showed that mechanical stimuli which activate Piezo1 result in a fluorescence increase of the GCaMP-G4 moiety of GenEpi whereas addition of ionomycin, which causes a global Ca rise in the cytoplasm, did not. They also showed that GCaMP-G4 (expressed free in the cytoplasm) co-expressed with Piezo-1, responded both to mechanical stimuli and to ionomycin. It is intriguing why the same Ca biosensor does not respond to a global cytosolic Ca rise when it is linked to Piezo1."

I understand the microdomain hypothesis (fig 1A). However, a global Ca rise by ionomycin should also affect GCaMP-G4 attached to Piezo1 on the membrane, since free cytoplasmic GCaMP-G4 is indeed sensitive to ionomycin. One potential reason that would explain the results shown in Fig 1C regarding GenEpi is that somehow the attachment of the Ca biosensor in GenEpi decreases its Ca affinity, such that a global Ca rise by ionomycin is not "sensed", compared to a much higher Ca microdomain at the Piezo1 channel mouth. This is speculative but, nevertheless, this was not the major point of concern.

"The authors show that GenEpi reports the mechanical activation of Piezo1. However, they do not show whether local Ca microdomains caused by other Ca-permeable channels affect the GenEpi response. Thus, the specificity claimed in the title is not sufficiently demonstrated... Piezo1 acts as a mechanosensor in many excitable cells (like those in Fig 4). In these cells, mechanical stimuli are associated with opening of voltage-activated calcium channels (like L-type channels, LTCCs), ... It is therefore important to check whether local Ca microdomains due to opening of LTCCs would contribute to the fluorescent readout of GenEpi, independent of the mechanical stimulus of Piezo1. This critical control seems to be lacking."

The authors responded to this: "As shown in the revised Fig. 4H-J, nifedipine treatment indeed abolished the beating of cardiomyocytes within the differentiated cardiac microtissues, suggesting that L-type channels were effectively blocked. Yet, the baseline fluorescence of the GenEpi-expressing cells (Fig. 4K, L) is not affected by blocking L-type channels."

Nifedipine blocks Ca influx through LTCCs and subsequent cardiomyocyte beating, so the results in new Fig 4H-L do not rule out that Ca influx through these channels affects GenEpi. Indeed, an interpretation of the same data is that part of the fluorescence readout of GenEpi in these contractile cells (without nifedipine) was due to Ca influx through LTCCs. This reviewer does not see what the "baseline fluorescence of the GenEpi-expressing cells" has to do with the question raised.

The best experiment was shown in the original (and revised) manuscript Fig 4C: blebbistatin (or other myosin inhibitors, such as BDM or para-aminoblebbistatin) uncouple contraction from excitation in cardiomyocytes: plasmalemma depolarization, opening of LTCCs, and Ca transients continue as in

control cells, but they are decoupled from contraction, so the mechanical stress on membrane Piezo1 channels should be largely eliminated. Indeed, in Fig 1C a representative experiment is shown in which blebbistatin abrogated the GenEPi response. However, as raised in the first review “the response of GenEPi was not completely abrogated in all cardiomyocytes since the frequency of beating merely decreased (Fig S16)” (current Fig S18). If the authors want to convincingly show that LTCCs in these excitable cells do not contribute to GenEPi fluorescence increase, they should aim to find conditions in which these myosin inhibitors completely stop cell motion. It is not sure this can be done, since often some motion remains with these inhibitors, and one could argue that this motion is sufficient to activate Piezo1 channels. But the nifedipine results shown in revised Fig 4H-L definitely do not clarify this critical point.

In non-excitable cells (or cells devoid of voltage-dependent Ca channels) as those shown in Fig 1 (HEK293T cells) GenEPi is probably reporting only Piezo-1 activity, so here the claim of the authors is OK. In excitable cells (Fig 4 and the zebrafish heart results in Fig 5), this is questionable. The conclusion in the rebuttal letter and revised manuscript “These data show that other calcium channels, such as L types channels, do not affect the Piezo1-dependent functional readout of GenEPi (lines 232-239)” is not supported by the data provided.

The further comment “In addition, we show in Supplementary Note 2 that the plasma membrane localization of GCaMP-G4 alone does not confer functional specificity to mechanical stimulation” is not relevant to the question raised. My point is that GCaMP attached to Piezo1 can be potentially sensitive to other Ca channels on the plasma membrane, not that Lck-GCaMP-G4 attached to the membrane can sense mechanical stimulation.

Second important point. From the first review “Fig 5 shows the response to a chemical activator of Piezo1, but it would add relevance to the results to show the response to a mechanical stimulus.” In the rebuttal letter the authors responded, “We demonstrate that GenEPi shows a highly stereotypical and mechanical stimuli dependent activation during zebrafish heart beating which is abolished when the heart is stopped with 2,3-Butanedione 2-monoxime (BDM) treatment (Fig. 5J-L).”

These results show the typical motion artifact observed when one images the moving zebrafish heart with a single wavelength Ca indicator such as GCaMP. The authors claim that the fluorescent changes originate from Ca influx through Piezo1 in GenEPi. However, Fig 5J and L suggest that the fluorescent changes are due to motion of the heart: the atrioventricular canal (AVC) moves in and out of a fixed region of interest (ROI) (I assume, because the ROI is not shown in Fig 5), as suggested by the large heart motion seen in Video 4. The figure should have shown the heart in systole and diastole, and thus the movement of the AVC in and out of the fixed ROI.

GCaMPs are never used in zebrafish heart without stopping the heart motion, see for instance (van Opbergen et al., 2018) because the fluorescence change due to motion is much larger than that due to the Ca rise. This is why the fluorescence in Fig 5J increased more than 3-fold (from less than 1 to more than 3 F/F0), about a six times larger change than the GenEPi response observed in HEK293 cells in Fig 1c. Indeed, treatment with the myosin inhibitor BDM stopped heart motion (and the motion artifact) (Fig 5K), although this was interpreted by the authors as just blocking the Piezo1 response, which probably did as well.

Unfortunately, single wavelength Ca indicators cannot be used in moving specimens. Only ratiometric biosensors have been shown to correct motion artifacts as those described above. Since GenEPi uses a single wavelength biosensor, this reviewer believes it cannot be used when the specimen movement is as large as that of the zebrafish embryo heart. Fig 5 I to L should be deleted.

Reference:

van Opbergen CJM, Koopman CD, Kok BJM, Knöpfel T, Renninger SL, Orger MB, Vos MA, van Veen TAB, Bakkers J & de Boer TP. (2018). Optogenetic sensors in the zebrafish heart: a novel in vivo electrophysiological tool to study cardiac arrhythmogenesis. *Theranostics* 8, 4750-4764.

REVIEWER COMMENTS

Reviewer #1 (Remarks to the Author)

With appropriate additional experimentation, the question has been answered. In the future, I hope to use such tools to clarify the contribution of piezo1 caused by different hemodynamic loads (e.g., left ventricular hypertrophy caused by systolic loading and cardiodilation caused by diastolic loading).

We are pleased to read that our “appropriate additional experimentation” has answered the questions raised by the reviewer and we share their excitement to use “such tools” to elucidate the role of Piezo1 in cardiac hypertrophy. We thank them for their constructive feedback during the review process.

Reviewer #2 (Remarks to the Author):

[Privately signs off in the 'Remarks to the Editor']

We are pleased to read that our responses to the comment's raised by the reviewer answered their initial questions. We thank them for their careful read of the original manuscript and their thorough and constructive suggestions to improve our work.

Reviewer #3 (Remarks to the Author):

The authors have added further results to address the questions raised in the review and have elucidated some points. However, some of the data and arguments proposed do not address and clarify the key criticisms:

From the first review “The authors showed that mechanical stimuli which activate Piezo1 result in a fluorescence increase of the GCaMP-G4 moiety of GenEPi whereas addition of ionomycin, which causes a global Ca rise in the cytoplasm, did not. They also showed that GCaMP-G4 (expressed free in the cytoplasm) co-expressed with Piezo-1, responded both to mechanical stimuli and to ionomycin. It is intriguing why the same Ca biosensor does not respond to a global cytosolic Ca rise when it is linked to Piezo1.”

I understand the microdomain hypothesis (fig 1A). However, a global Ca rise by ionomycin should also affect GCaMP-G4 attached to Piezo1 on the membrane, since free cytoplasmic GCaMP-G4 is indeed sensitive to ionomycin. One potential reason that would explain the results shown in Fig 1C regarding GenEPi is that somehow the attachment of the Ca biosensor in GenEPi decreases its Ca affinity, such that a global Ca rise by ionomycin is not “sensed”, compared to a much higher Ca microdomain at the Piezo1 channel mouth. This is speculative but, nevertheless, this was not the major point of concern.

We thank the reviewer for sharing their deep thinking on the working mechanism of GenEPi. We show in Supplementary Fig. 4 (initial and revised manuscript) that the calcium affinity of GCaMP-G4 within GenEPi is not affected by the fusion and retains its low affinity for calcium. As we show in Fig. 1C, the duplication of the flexible linker within Piezo1-2xGSGG-GCaMP-G4 (2xGSGG in contrast to 1xGSGG in GenEPi) is sufficient to permit GCaMP-G4 to sense the ionomycin-triggered global calcium increase. Our observations align with previous work on the nature of calcium microdomains which provide means for signaling specificity (Parekh, 2008). Consequently, our current hypothesis is that calcium microdomains of Piezo1 are spatially protected in order to permit specific cellular responses to occur without compromising other signaling cascades. Indeed, our TIRFM analysis demonstrates that calcium increases can be reported for individual (spatially distinct) clusters without affecting the activity of nearby ones (~1 μ m distance) (Fig. 3I, I2 and I3, J and K). We anticipate that future work using GenEPi and similar biosensors will allow for acquiring a fundamental understanding of the nature of calcium microdomains on cell membranes.

“The authors show that GenEPi reports the mechanical activation of Piezo1. However, they do not show whether local Ca microdomains caused by other Ca-permeable channels affect the GenEPi response. Thus, the specificity claimed in the title is not sufficiently demonstrated... Piezo1 acts as a mechanosensor in many excitable cells (like those in Fig 4). In these cells, mechanical stimuli are associated with opening of voltage-activated calcium channels (like L-type channels, LTCCs), ... It is therefore important to check whether local Ca microdomains due to opening of LTCCs would contribute to the fluorescent readout of GenEPi, independent of the mechanical stimulus of Piezo1. This critical control seems to be lacking.”

The authors responded to this: “As shown in the revised Fig. 4H-J, nifedipine treatment indeed abolished the beating of cardiomyocytes within the differentiated cardiac microtissues, suggesting that L-type channels were effectively blocked. Yet, the baseline fluorescence of the GenEPi-expressing cells (Fig. 4K, L) is not affected by blocking L-type channels.”

Nifedipine blocks Ca influx through LTCCs and subsequent cardiomyocyte beating, so the results in new Fig 4H-L do not rule out that Ca influx through these channels affects GenEPi. Indeed, an interpretation of the same data is that part of the fluorescence readout of GenEPi in these contractile cells (without nifedipine) was due to Ca influx through LTCCs. This reviewer does not see what the “baseline fluorescence of the GenEPi-expressing cells” has to do with the question raised.

The best experiment was shown in the original (and revised) manuscript Fig 4C: blebbistatin (or other myosin inhibitors, such as BDM or para-aminoblebbistatin) uncouple contraction from excitation in cardiomyocytes: plasmalemma depolarization, opening of LTCCs, and Ca transients continue as in control cells, but they are

decoupled from contraction, so the mechanical stress on membrane Piezo1 channels should be largely eliminated. Indeed, in Fig 1C a representative experiment is shown in which blebbistatin abrogated the GenEPi response. However, as raised in the first review “the response of GenEPi was not completely abrogated in all cardiomyocytes since the frequency of beating merely decreased (Fig S16)” (current Fig S18). If the authors want to convincingly show that LTCCs in these excitable cells do not contribute to GenEPi fluorescence increase, they should aim to find conditions in which these myosin inhibitors completely stop cell motion. It is not sure this can be done, since often some motion remains with these inhibitors, and one could argue that this motion is sufficient to activate Piezo1 channels. But the nifedipine results shown in revised Fig 4H-L definitely do not clarify this critical point.

In non-excitable cells (or cells devoid of voltage-dependent Ca channels) as those shown in Fig 1 (HEK293T cells) GenEPi is probably reporting only Piezo-1 activity, so here the claim of the authors is OK. In excitable cells (Fig 4 and the zebrafish heart results in Fig 5), this is questionable. The conclusion in the rebuttal letter and revised manuscript “These data show that other calcium channels, such as L types channels, do not affect the Piezo1-dependent functional readout of GenEPi (lines 232-239)” is not supported by the data provided.

The further comment “In addition, we show in Supplementary Note 2 that the plasma membrane localization of GCaMP-G4 alone does not confer functional specificity to mechanical stimulation” is not relevant to the question raised. My point is that GCaMP attached to Piezo1 can be potentially sensitive to other Ca channels on the plasma membrane, not that Lck-GCaMP-G4 attached to the membrane can sense mechanical stimulation.

We thank the reviewer for their comment. To investigate whether GenEPi signal is influenced by other calcium permeated ion channels, we tested in our original manuscript its response to ATP-induced calcium release. Such global calcium releases depend on P2X receptors which have similar conductance to Piezo1 (P2X ~20pS and Piezo1 ~25pS) (Evans, 1996, Coste et al., 2010). As we show in Supplementary Fig. 8, GenEPi does not respond to ATP-induced calcium fluctuations which demonstrates that the activation of P2X channels does not affect its fluorescence response.

In our revised manuscript, we investigated whether L-type channels can influence GenEPi response. As shown in the revised Fig. 4H-J, nifedipine treatment abolished the beating of cardiomyocytes within the differentiated cardiac microtissues, suggesting that L-type channels were effectively blocked. However, the blockade of the L-type channels did not affect the baseline activity fluorescence of GenEPi (e.g. activation due to intrinsic cellular forces). Indeed, its baseline activity (F_0) just before the contraction is the same with the fluorescence activity after effective blockade of the L-type channels. Given the nature of calcium microdomains on cell membranes (see previous response) and the fact that the conductance of L-type channels (~2.4pS) is around one order of magnitude smaller (Church and Stanley, 1996) than that of

Piezo1 and P2X channels, we conclude that GenEpi provides a Piezo1-specific readout which is not influenced by the activation of other calcium channels.

Finally, as shown in Supplementary Note 2-Fig.1 the membrane localization of GCaMP-G4 alone is not sufficient to provide functional specificity; targeting of the GCaMP to the ion channel calcium microdomain is needed to report specific channel calcium events. Thus, our design principle of GenEpi can serve as a blueprint for developing and engineering optical reporters of other ion channels without affecting their function.

Second important point. From the first review “Fig 5 shows the response to a chemical activator of Piezo1, but it would add relevance to the results to show the response to a mechanical stimulus.” In the rebuttal letter the authors responded, “We demonstrate that GenEpi shows a highly stereotypical and mechanical stimuli dependent activation during zebrafish heart beating which is abolished when the heart is stopped with 2,3-Butanedione 2-monoxime (BDM) treatment (Fig. 5J-L).”

These results show the typical motion artifact observed when one images the moving zebrafish heart with a single wavelength Ca indicator such as GCaMP. The authors claim that the fluorescent changes originate from Ca influx through Piezo1 in GenEpi. However, Fig 5J and L suggest that the fluorescent changes are due to motion of the heart: the atrioventricular canal (AVC) moves in and out of a fixed region of interest (ROI) (I assume, because the ROI is not shown in Fig 5), as suggested by the large heart motion seen in Video 4. The figure should have shown the heart in systole and diastole, and thus the movement of the AVC in and out of the fixed ROI.

GCaMPs are never used in zebrafish heart without stopping the heart motion, see for instance (van Opbergen et al., 2018) because the fluorescence change due to motion is much larger than that due to the Ca rise. This is why the fluorescence in Fig 5J increased more than 3-fold (from less than 1 to more than 3 F/F₀), about a six times larger change than the GenEpi response observed in HEK293 cells in Fig 1c. Indeed, treatment with the myosin inhibitor BDM stopped heart motion (and the motion artifact) (Fig 5K), although this was interpreted by the authors as just blocking the Piezo1 response, which probably did as well.

Unfortunately, single wavelength Ca indicators cannot be used in moving specimens. Only ratiometric biosensors have been shown to correct motion artifacts as those described above. Since GenEpi uses a single wavelength biosensor, this reviewer believes it cannot be used when the specimen movement is as large as that of the zebrafish embryo heart. Fig 5 I to L should be deleted.

*Reference: van Opbergen CJM, Koopman CD, Kok BJM, Knöpfel T, Renninger SL, Orger MB, Vos MA, van Veen TAB, Bakkens J & de Boer TP. (2018). Optogenetic sensors in the zebrafish heart: a novel in vivo electrophysiological tool to study cardiac arrhythmogenesis. *Theranostics* 8, 4750-4764.*

We thank the reviewer for their comment. Single channel indicators have been previously used to monitor calcium signaling in zebrafish heart with or without normalization approaches (Juan et al., 2023, Fukui et al., 2021)). In our revised Fig. 5, we have normalized the GenEPi signal acquired from dynamic widefield time-lapse imaging of *Tg(hsp70:GenEPi)* zebrafish hearts to the typical motion artifact signal of *Tg(kdrl:NLS-mCherry)* zebrafish beating hearts. We have now added this information to our revised manuscript (e.g. lines 288-290, 1169-1173, Supplementary Fig. 22). Note that the mean difference of the average F/F_0 of GenEPi after and before normalization is less than 5% (Revision Fig. 2.1).

Revision Fig. 2.1. Difference of zebrafish heart GenEPi responses before and after normalization to the *kdrl:NLS-mCherry* signal and value distribution. Note that the mean of the difference is less than -5% ($-3.802 \pm 0.8105\%$, Mean \pm SEM).

References

- CHURCH, P. J. & STANLEY, E. F. 1996. Single L-type calcium channel conductance with physiological levels of calcium in chick ciliary ganglion neurons. *J Physiol*, 496 (Pt 1), 59-68.
- COSTE, B., MATHUR, J., SCHMIDT, M., EARLEY, T. J., RANADE, S., PETRUS, M. J., DUBIN, A. E. & PATAPOUTIAN, A. 2010. Piezo1 and Piezo2 are essential components of distinct mechanically activated cation channels. *Science*, 330, 55-60.
- EVANS, R. J. 1996. Single channel properties of ATP-gated cation channels (P2X receptors) heterologously expressed in Chinese hamster ovary cells. *Neurosci Lett*, 212, 212-4.
- FUKUI, H., CHOW, R. W., XIE, J., FOO, Y. Y., YAP, C. H., MINC, N., MOCHIZUKI, N. & VERMOT, J. 2021. Bioelectric signaling and the control of cardiac cell identity in response to mechanical forces. *Science*, 374, 351-354.
- JUAN, T., RIBEIRO DA SILVA, A., CARDOSO, B., LIM, S., CHARTEAU, V. & STAINIER, D. Y. R. 2023. Multiple *pkd* and *piezo* gene family members are required for atrioventricular valve formation. *Nat Commun*, 14, 214.
- PAREKH, A. B. 2008. Ca^{2+} microdomains near plasma membrane Ca^{2+} channels: impact on cell function. *J Physiol*, 586, 3043-54.

REVIEWERS' COMMENTS

Reviewer #3 (Remarks to the Author):

The authors, with additional data, have answered the comments raised by this reviewer. They show that other local calcium microdomains caused by membrane channels (ATP-gated channels, P2x receptors) do not affect the readout of GenEPi (Fig. S8).

In addition, in the revised Fig 5, they normalized the GenEPi signal to that of NLS-mCherry, which is not sensitive to calcium but affected by motion in the beating heart. Thus, the corrected GenEPi signal in the atrioventricular canal (Fig 5L) seems to be due to activation of Piezo-1.